# NUDT2 initiates viral RNA degradation by removal of 5′-phosphates

Beatrice T. Laudenbach[1,2,14], Karsten Krey [1,14], Quirin Emslander[1,14], Line Lykke Andersen[1], Alexander Reim [3], Pietro Scaturro [1,4], Sarah Mundigl[1], Christopher Dächert[5,6], Katrin Manske[7], Markus Moser[8,9], Janos Ludwig[10], Dirk Wohlleber [7], Andrea Kröger[11,12], Marco Binder [5] & Andreas Pichlmair [1,2,13 ✉]

While viral replication processes are largely understood, comparably little is known on cellular mechanisms degrading viral RNA. Some viral RNAs bear a 5′-triphosphate (PPP-) group that impairs degradation by the canonical 5′-3′ degradation pathway. Here we show that the Nudix hydrolase 2 (NUDT2) trims viral PPP-RNA into monophosphorylated (P)-RNA, which serves as a substrate for the 5′-3′ exonuclease XRN1. NUDT2 removes 5′-phosphates from PPP-RNA in an RNA sequence- and overhang-independent manner and its ablation in cells increases growth of PPP-RNA viruses, suggesting an involvement in antiviral immunity. NUDT2 is highly homologous to bacterial RNA pyrophosphatase H (RppH), a protein involved in the metabolism of bacterial mRNA, which is 5′-tri- or diphosphorylated. Our results show a conserved function between bacterial RppH and mammalian NUDT2, indicating that the function may have adapted from a protein responsible for RNA turnover in bacteria into a protein involved in the immune defense in mammals.

[1] Technical University of Munich, School of Medicine, Institute of Virology, 81675 Munich, Germany. [2] Innate Immunity Laboratory, Max-Planck Institute of Biochemistry, Martinsried/Munich, Germany. [3] Department of Proteomics and Signal transduction, Max-Planck Institute of Biochemistry, Martinsried/Munich, Germany. [4] Leibniz Institute for Experimental Virology (HPI), Hamburg, Germany. [5] Research Group "Dynamics of Early Viral Infection and the Innate Antiviral Response" (division F170), German Cancer Research Center, Heidelberg (DKFZ), Heidelberg, Germany. [6] Faculty of Biosciences, Heidelberg University, 69120 Heidelberg, Germany. [7] Technical University of Munich, School of Medicine, Institute of Molecular Immunology, Munich, Germany. [8] Department of Molecular Medicine, Max-Planck Institute of Biochemistry, Martinsried/Munich, Germany. [9] Technical University of Munich, School of Medicine, Institute of Experimental Hematology, Munich, Germany. [10] Institute of Clinical Chemistry and Clinical Pharmacology, University Hospital Bonn, Bonn, Germany. [11] Otto von Guericke University Magdeburg, Institute for Medical Microbiology, Magdeburg, Germany. [12] Helmholtz Centre for Infection Research, Innate Immunity and Infection, Braunschweig, Germany. [13] German Center for Infection Research (DZIF), Munich partner site, Munich, Germany. [14]These authors contributed equally: Beatrice T. Laudenbach, Karsten Krey, Quirin Emslander. ✉email: andreas.pichlmair@tum.de

All living organisms use RNA as the messenger molecule to translate their DNA encoded genetic information into proteins. Besides trans-activating sequences encoded on messenger (m)RNA and the subcellular localization of the RNA molecule, the abundance of mRNA is the primary determinant to modulate gene expression. mRNA abundance is regulated based on transcription and the stability of the molecule. In eukaryotes, the most prominent and evolutionary conserved processes that modulate the stability of mRNA are 5′-to-3′ RNA degradation by the cellular exonuclease 1 (XRN1) and 3′-to-5′ degradation by the RNA exosome complex. Eukaryotic mRNAs are protected from 5′-to-3′ degradation by a 5′-m7G cap structure, co-transcriptionally added to the mRNA molecule. To proceed with the 5′-to-3′ degradation process, the mRNA cap needs to be removed by the decapping protein 2 (DCP2) to generate a monophosphorylated (P-)RNA substrate, which can then be degraded by XRN1 (Fig. 1a)[1,2]. Genomic RNAs and RNA transcripts of many viruses, including Orthomyxo-, Arena-, Paramyxo- and Bunyaviruses, do not bear a 5′-cap but present a 5′-triphosphate (PPP-) group. This terminal PPP- moiety linked to double-stranded RNA serves as a pathogen-associated molecular pattern (PAMP) sensed by the cellular helicase RIG-I, which activates a MAVS-dependent innate immune signaling cascade. Eventually, this culminates in the accumulation of type-I interferon (IFN) and IFN-stimulated genes and is critical to prevent virus spread[3]. It is currently unclear why some viruses bear terminal triphosphates despite the selective pressure of the innate immune system. However, a possible explanation may be the increased stability of PPP-RNA that inhibits 5′-to-3′ degradation by XRN1. Interestingly, despite the stabilizing PPP-group on viral RNA, specific highly secondary structured viral RNA sequences expressed by Flavi-, Bunya-, and Arenaviruses impair the activity of XRN1[4,5]. Moreover, the depletion of XRN1 increases the stability of viral RNA[6]. Collectively, this suggests that 5′-to-3′ RNA degradation processes are operative in the case of viral RNA. In analogy to the eukaryotic mRNA degradation system, however, the degradation of viral PPP-RNA would require a processing step that removes the PPP-group. Indeed, recently the

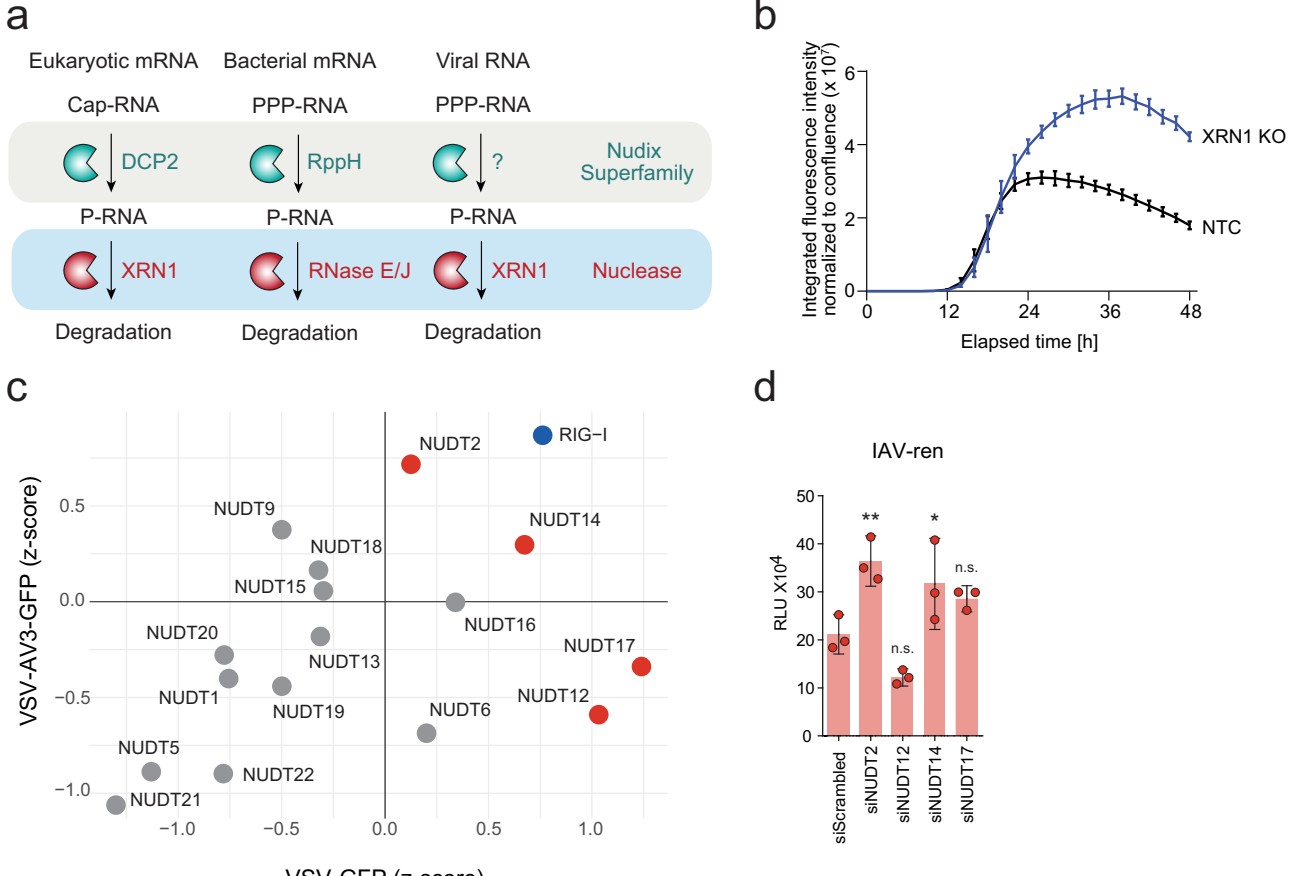

**Fig. 1 Influence of Nudix depletion on virus growth. a** Schematic representation of processes involved in 5′-to-3′ RNA degradation. Eukaryotic and bacterial triphosphorylated messenger (m)RNA are processed by proteins DCP2 and RppH, respectively, to generate monophosphorylated substrates for the 5′-3′ exonuclease XRN1, RNase E or RNase J. We hypothesize that a mammalian Nudix hydrolase could prepare viral RNA for degradation by removing pyrophosphate groups. **b** Hep3B cells with a CRISPR/Cas9-mediated KO of XRN1 or Hep3B cells transduced with a non-targeting-control vector (NTC) were infected with VSV-GFP at MOI 0.01 for 48 h. GFP expression was continuously quantified by live-cell microscopy. The data points are displayed as the mean of five technical replicates ± SD. One representative experiment of three is shown. **c** HeLa cells transfected with siRNA pools targeting the indicated Nudix gene for 48 h were infected with VSV-GFP (x-axis) or VSV-AV3-GFP (y-axis). Twenty-four hours after infection, GFP expression was analyzed by fluorometric analysis. The displayed z-scores were calculated from three independent reactions. **d** HeLa cells were treated with siRNA pools for 48 h targeting the transcript of *NUDT2, NUDT12, NUDT14, NUDT17*, or non-targeting siRNA control (siScrambled) and infected with Influenza A reporter virus expressing the renilla luciferase (IAV-ren) at MOI 1. Forty-eight hours post-infection, renilla luciferase activity was analyzed. The means of three biological replicates ± SD are shown; ** $p = 0.004$, * $p = 0.03$ as analyzed by two-way analysis of variance (ANOVA) statistics. RLU relative light units. n.s. not significant.

triphosphatases RNA/RNP complex-1-interacting phosphatase (DUSP11) and Decapping and exoribonuclease protein (DXO, also known as DOM3Z) were shown to dephosphorylate PPP-RNA to P-RNA that can be further degraded by XRN1[7–9]. However, the mechanism of PPP-RNA dephosphorylation and the substrate specificity of DUSP11 is not yet well understood.

Similar to viral RNA, prokaryotic mRNA also bears a 5′-PPP or 5′-PP group[10,11]. In bacteria, the 5′ phosphates can be removed by the RNA pyrophosphohydrolase (RppH) to prepare RNA for degradation by the bacterial 5′-to-3′ endoribonuclease RNase E or 5′ exonucleases (e.g., RNase J[11–13] (Fig. 1a)). Interestingly, both mammalian DCP2 and bacterial RppH belong to the superfamily of Nudix hydrolases[14], a protein family that is highly diverse in terms of sequence, domain organization, and substrate specificity[15]. Nudix proteins are characterized by a Nudix hydrolase domain, which contains characteristic catalytic and metal-binding amino acids[16]. They mostly function as pyrophosphohydrolases active towards substrates with the structure NDP-X (nucleoside diphosphate linked to another moiety, X), usually resulting in NMP (nucleoside monophosphate) and P-X as products[16–18]. Since Nudix hydrolases process mammalian and bacterial mRNAs, we tested whether mammalian Nudix hydrolases may have dephosphorylation activity on PPP-RNA to initiate further exonucleolytic processing. Understanding such viral RNA degradation mechanisms is essential to further study cellular antiviral principles that are in place to limit the spread and pathogenicity of infectious pathogens and their involvement in antiviral immunity.

## Results

**Nudix hydrolases have an impact on virus growth.** To establish a screening system that allows testing for 5′-to-3′ dependent RNA degradation in virus-infected cells, we first evaluated whether CRISPR/Cas9-mediated depletion of XRN1 affects the replication of PPP-RNA-generating vesicular stomatitis virus expressing green fluorescent protein (VSV-GFP). Automated live-cell microscopy of VSV-GFP infected cells showed a distinct increase of GFP signal in XRN1 targeted cells as compared to controls (Fig. 1b), indicating that VSV-GFP is restricted by a 5′-to-3′ exonuclease dependent pathway. We hypothesized that any Nudix hydrolase that may be involved in PPP-RNA processing should similarly negatively regulate virus growth. We, therefore, used siRNAs to deplete mammalian Nudix proteins, which are known to play a role in nucleotide metabolism[17] (Supplementary Table 1) and quantified the viral growth of VSV-GFP and the IFN-stimulating variant VSV-AV3-GFP[19]. Compared to the control treatment, depletion of the PPP-RNA-sensing pattern recognition receptor RIG-I led to increased GFP expression after infection with either of the two viruses, confirming the validity of this approach (Fig. 1c). Notably, the depletion of four Nudix hydrolases (NUDT2, NUDT12, NUDT14, and NUDT17) led to an increase in VSV-GFP or VSV-AV3-GFP growth. As tested by cell titer glow assay, cell viability was not affected by the depletion of these Nudix proteins (Supplementary Fig. 1a). Depletion of NUDT5, NUDT19, NUDT21, and NUDT22 showed a negative impact on VSV growth (Fig. 1c). For NUDT19 and NUDT22, this phenotype could potentially be explained by reduced cell viability after depletion of these Nudix hydrolases. In total, we selected four Nudix hydrolases (NUDT2, -12, -14, and -17), which showed the highest suppressive effect on either of the two viruses for further experiments. We confirmed the knockdown efficiency of NUDT2, -12, -14, and -17 siRNA treatment in HeLa cells by RT-qPCR and western blotting (NUDT2) (Supplementary Fig. 1b, c). We used renilla-luciferase-expressing Influenza A virus (IAV-ren) as an alternative PPP-RNA generating virus to confirm the

antiviral activity of the four NUDTs (Fig. 1d). While the depletion of NUDT12 and NUDT17 did not significantly increase the renilla signal in this cell type, the depletion of NUDT2 and -14 led to a significant increase (Fig. 1d).

**NUDT2 releases phosphates from a triphosphorylated RNA substrate.** To characterize the four selected Nudix hydrolases for their potential ability to dephosphorylate PPP-RNA, we generated recombinant NUDT2, -12, -14, and -17 in *E. coli*. As controls, we mutated the metal-coordinating glutamic acid residue (E) in their conserved Nudix hydrolase domain motif (NUDT2 E58A, NUDT12 E369A, NUDT14 E121A, and NUDT17 E124A)[20]. Successful purification of the recombinant proteins was confirmed by SDS-PAGE and mass spectrometry (Supplementary Fig. 2a, b). As expected, co-incubation with in vitro transcribed RNAs showed that the purified proteins and recombinant calf intestinal phosphatase (CIP) did not have RNase activity (Fig. 2a). RNase A-treated RNA, however, was clearly degraded. We next tested whether the recombinant Nudix proteins can dephosphorylate PPP-RNA. For this, we used thin-layer chromatography (TLC), which allows visualization of released γ-phosphates from γ-$^{32}$P-labeled single-stranded 44-mer PPP-RNA substrate. Notably, NUDT2 released substantial amounts of γ-phosphate (Fig. 2b, γ-$^{32}$P) and led to almost complete dephosphorylation of the substrate RNA (Fig. 2b, RNA). In contrast, the other tested Nudix proteins (NUDT12, NUDT14, and NUDT17) only showed a minute release of γ-$^{32}$P. The NUDT2-dependent dephosphorylation was time-dependent (Fig. 2c) and required Nudix hydrolase activity since the NUDT2 E58A mutant did not release radioactively labeled phosphates or change RNA substrate labeling (Fig. 2d). Dephosphorylation was also apparent when using double-stranded RNA substrate, which was generated by annealing antisense RNA to the radioactively labeled 44-mer substrate (dsRNA) and by using radioactively labeled IVT4 (hairpin RNA) as substrate (Fig. 2e). Bacterial RppH, known for its activity to release γ-phosphates from unpaired RNA 5′-overhangs[11], showed a similar, although less pronounced release of phosphates from single-stranded PPP-RNA substrates (Fig. 2e). These experiments indicated that human NUDT2, like bacterial RppH, has dephosphorylation activity, pointing towards NUDT2 as a candidate protein involved in PPP-RNA degradation and antiviral immunity.

Interestingly, sequence alignment analysis suggests that NUDT2 is highly conserved in evolutionary distant eukaryotic species (Supplementary Fig. 2c). Notably, after sequence alignment of all human Nudix hydrolases, NUDT2 shows the highest sequence similarity to bacterial RppH, further pointing towards a conserved function of both proteins (Fig. 2f). The crystal structures of RppH and NUDT2 have been solved[20,21], and alignment of these structures revealed striking similarities in which the active site of NUDT2 resembles the RNA binding pocket of RppH (Fig. 2g). It exhibits a negatively charged interface required for positioning the positively charged magnesium ions needed to coordinate the RNA's PPP-group. Furthermore, the RNA binding pocket contains a large, positively charged channel bearing space for the negatively charged RNA phosphate backbone (Fig. 2h).

**NUDT2 is active on a broad range of PPP-RNA substrates.** We focused on NUDT2 as a candidate protein involved in viral RNA degradation and therefore assessed the ability of NUDT2 to release phosphates from different substrates. To this aim, we used a time-resolved enzymatic assay to compare phosphate release from PPP-RNA. Titration of NUDT2 suggested an optimal signal-to-noise ratio when 600 nM of NUDT2 were used

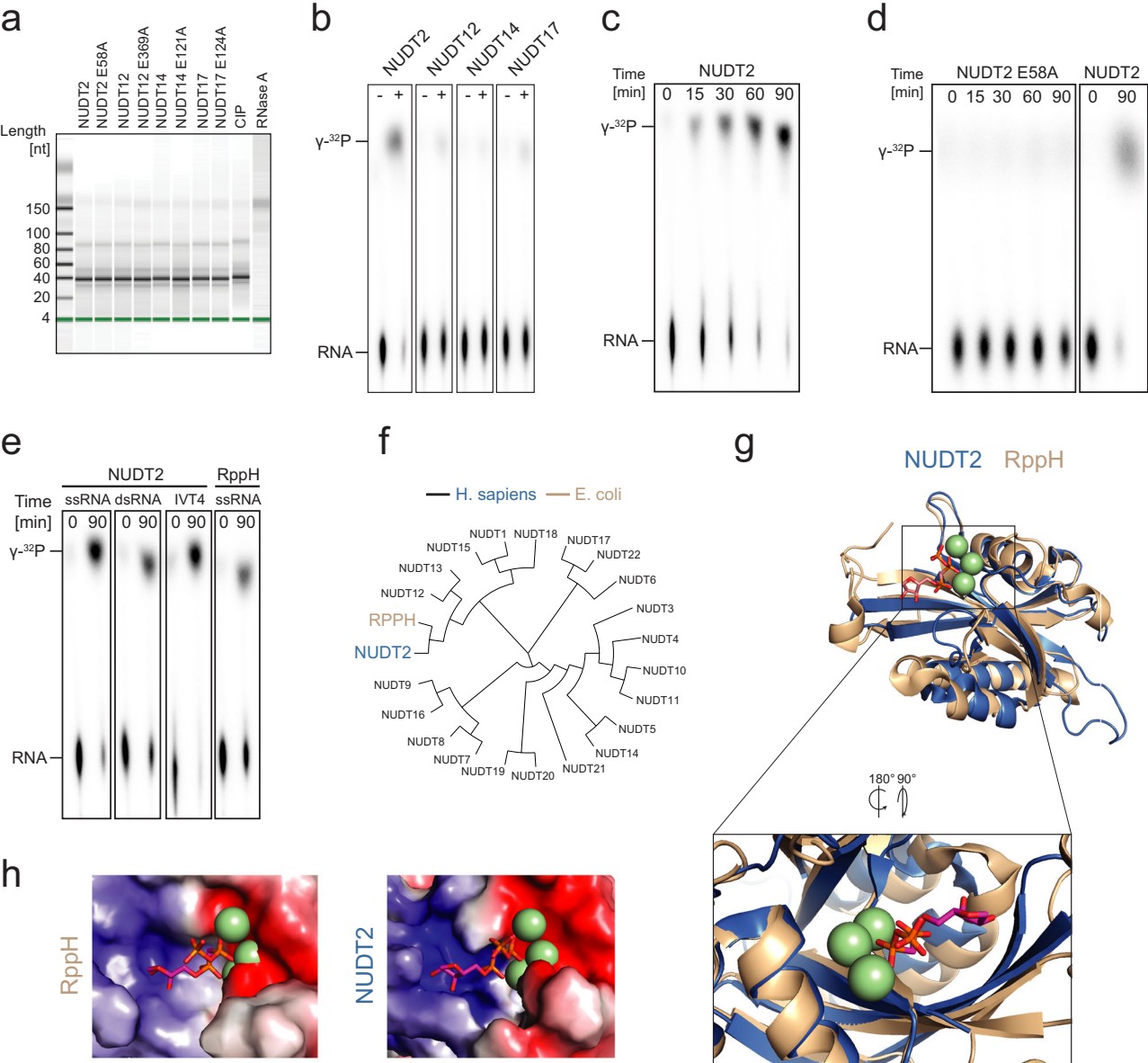

**Fig. 2 Identification of NUDT2 as PPP-RNA phosphatase. a** RNA was incubated for 3 h with the indicated Nudix hydrolase, and the integrity of the RNA was determined using a Bioanalyzer. One representative experiment of three is shown. **b** γ-$^{32}$P-radiolabeled in vitro transcribed 44-mer ssRNA substrate was left untreated or incubated for 90 min with 600 nM NUDT2, NUDT12, NUDT14 or NUDT17 (+) or was left untreated (−). Reaction mixtures were analyzed by TLC and autoradiography. γ-$^{32}$P: radiolabeled γ-phosphate; RNA: remaining input RNA after incubation. **c** As (**b**) but incubation for the time indicated at the top of the panel. 600 nM NUDT2 were used. **d** As (**c**) but 600 nM NUDT2 E58A or NUDT2 were used. **e** As (**b**) but γ-$^{32}$P-radiolabeled in vitro transcribed single-stranded (ss-) or double-stranded (ds-) RNA generated by annealing an antisense oligo, or a RIG-I activating hairpin RNA (IVT4) were incubated with 600 nM NUDT2 and 5 U RppH, respectively. **f** Phylogenetic analysis of all human Nudix hydrolases and bacterial RppH was performed using MAFFT. **g** Structural alignment of the crystal structures of RppH (brown) with an RNA substrate (red sticks) (PDB: 4S2Y) and NUDT2 (blue) (PDB: 3U53) using PYMOL[57]. The green-colored magnesium ions spheres are essential for hydrolase activity. **h** Visualization of the electrostatic surface charges of RppH and NUDT2 displayed with the aligned RNA of the RppH structure using PYMOL. Acidic amino acids are depicted in red, and basic residues are shown in blue.

(Supplementary Fig. 3a). Phosphatase activity of CIP was higher, probably due to more efficient or more complete dephosphorylation of the RNA substrate. As expected, NUDT2 mediated phosphate release is dependent on a glutamic acid at the catalytic site since the NUDT2 E58A mutant or control proteins (NUDT14, NUDT14 E121A) failed to yield a considerable signal (Fig. 3a).

To investigate the reactivity of NUDT2 on different substrates, we designed RNAs with or without different 5′ single-stranded overhangs and variable nucleotides at the first (A, G) and the second position (A, G, U, and C) of the RNA (Fig. 3b). NUDT2 processed all constructs, irrespective of the length of the 5′ single-stranded overhang (Fig. 3c). Moreover, NUDT2 was also able to dephosphorylate RNA substrates with adenines at the 5′ end and 5′ single-stranded RNA overhang starting with adenines (Fig. 3c). As expected, the control NUDT2 E58A did not mediate significant phosphate release (Supplementary Fig. 3b). NUDT2 did not degrade the RNA substrates (Fig. 3d), confirming the lack of nuclease activity of NUDT2. Importantly, NUDT2 did not release any phosphates from single nucleotides (rATP, rGTP,

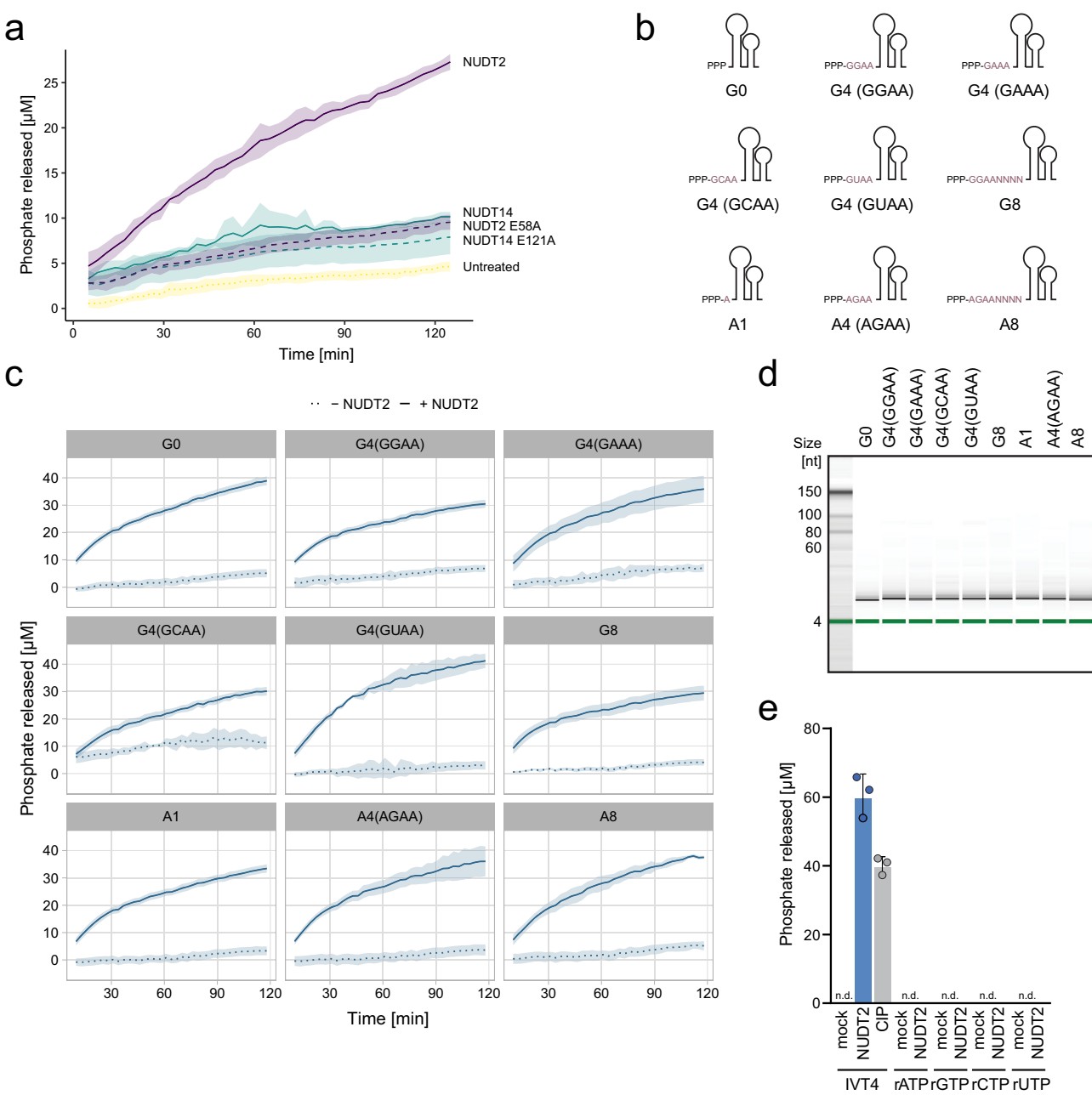

**Fig. 3 NUDT2 is active on a broad range of substrates. a** EnzChek assay to quantify released phosphate from in vitro transcribed double-stranded triphosphorylated hairpin RNA (IVT4) left untreated (untreated) or incubated with recombinant NUDT2, NUDT E58A, NUDT14, or NUDT14 E121A (600 nM), as indicated. Colorimetric measurements of the samples were performed every 3 min. The line graph shows the mean of three independent experiments with the ribbon indicating ± SD. **b** Illustration of in vitro transcribed RNA substrates with varying starting nucleotides with or without paired 5′-overhangs. **c** RNA substrates depicted in (**b**) were incubated without (dotted line) or with (solid line) NUDT2 at a concentration of 600 nM over a time course of 2 h, and phosphate release was evaluated using the EnzChek assay. Spectrophotometric measurements to quantify phosphate release were performed every 3 min. The line graph shows the mean of three independent experiments with the ribbon indicating ± SD. **d** The indicated RNA was incubated with NUDT2 (600 nM) for 3 h, and the integrity was analyzed using an Agilent small RNA chip and a Bioanalyzer. **e** The indicated ribonucleoside triphosphates (2.5 μM) and the positive control IVT4 were incubated for 3 h with NUDT2 (600 nM), and phosphate release was determined by malachite green assay. The mean of three independent reactions ± SD is shown.

rUTP, and rCTP) at comparable molarity (Fig. 3e), indicating that NUDT2 prefers polynucleotides as substrates. We concluded from these results, that NUDT2 is dephosphorylating PPP-RNAs irrespective of the first nucleotide (A or G), the 5′-terminus sequence, and the base-pairing of the 5′-end (single/double-stranded) of the substrate. Regarding substrate preference, NUDT2 appears to be different from RppH, which requires an unpaired 5′ overhang for its activity[11] (Supplementary Fig. 3c).

**NUDT2 releases monophosphates in a sequential manner**. To assess the mode of phosphate removal by NUDT2, we synthesized PPP-dinucleotides (PPP-GpA and PPP-ApG) and tested how NUDT2 processes these substrates. Notably, NUDT2 sequentially released single phosphates resulting in a di-phosphorylated dinucleotide intermediate that is further converted into a mono-phosphorylated product (Fig. 4a–c). This contrasts with RppH, which predominantly releases

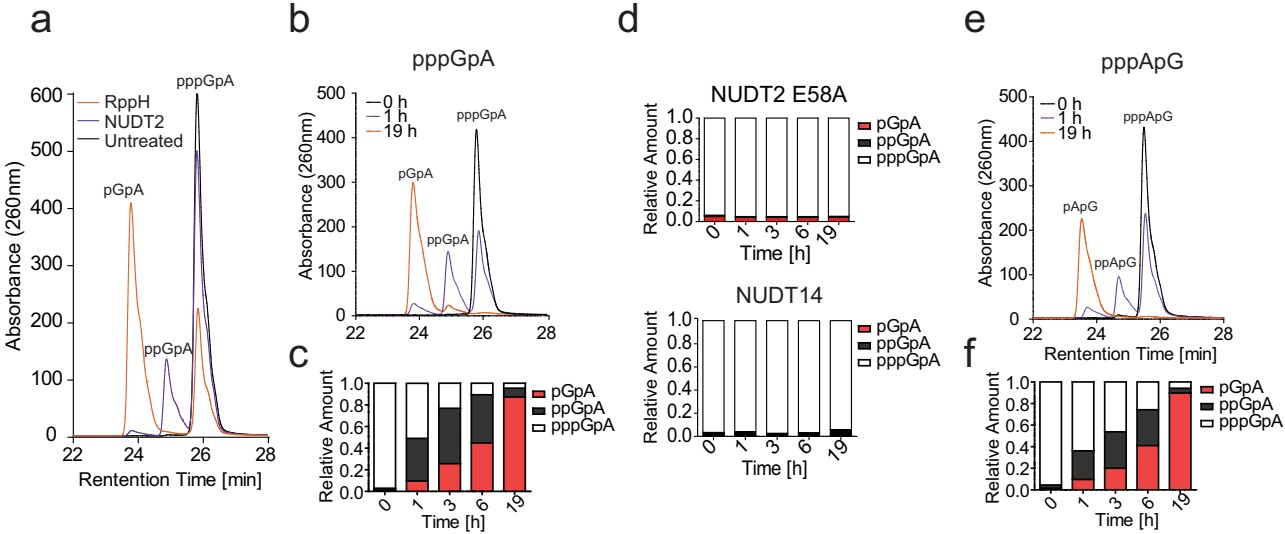

**Fig. 4 NUDT2 sequentially releases phosphates from PPP-RNA. a** A triphosphorylated RNA dinucleotide (pppGpA) was treated with NUDT2 (600 nM) or RppH (5 U) for 2 h at 37 °C. Additionally, the reference pppGpA was incubated in the same reaction buffer (untreated). Products were analyzed by reverse-phase HPLC. The absorbance at 260 nm at the indicated retention time is shown. **b** As (**a**) incubation with 600 nM NUDT2 for the indicated time. **c** As (**b**) but quantification of the generated RNA after incubation of pppGpA with NUDT2 for the indicated time. Quantification was performed by the integration of the corresponding peak area. **d** As (**b**) but incubation with NUDT2 E58A or NUDT14. **e** As (**b**), but the triphosphorylated dinucleotide pppApG was used as substrate. **f** As (**e**) but quantification of the generated RNA after incubation of pppApA with NUDT2 for the indicated time. Quantification was performed by the integration of the corresponding peak area.

pyrophosphates from PPP-GpA in a one-step reaction without detectable PP-RNA intermediate[22] (Fig. 4a). Neither NUDT2 E58A mutant nor NUDT14 changed the phosphorylation status of PPP-GpA, confirming the activity of NUDT2 in this assay (Fig. 4d). Removal of monophosphates by NUDT2 was also observed when using a PPP-ApG nucleotide as substrate, further corroborating that RNAs containing guanosines or adenosines at the 5′ end are similarly dephosphorylated by NUDT2 (Figs. 3c, 4e, f).

**NUDT2 and XRN1 cooperate for PPP-RNA degradation.** We next tested whether NUDT2 can prepare PPP-RNA for degradation by the canonical XRN1 dependent 5′-to-3′ RNA degradation machinery. As a substrate, we used Hepatitis C virus (HCV) encoding PPP-RNA. Incubation with NUDT2 or XRN1 did not affect the stability of the PPP-HCV RNA (Fig. 5a, b), indicating that the PPP-group protects RNA from degradation by XRN1 and that NUDT2 does not have exonuclease activity. Notably, co-incubation of HCV PPP-RNA with NUDT2 and XRN1 led to RNA degradation, confirming that the two proteins can act in concert to degrade PPP-RNA (Fig. 5a, b). Control experiments using bacterial RppH, as well as RppH/XRN1 co-treatment gave similar results. These data demonstrate the functional synergy between NUDT2 and XRN1 in a 5′-to-3′ degradation pathway of PPP-RNA. As NUDT2 consecutively dephosphorylates PPP-RNA (Fig. 4a, b, e), this suggests that mammalian NUDT2 can, in principle, process PPP-RNA to prepare it for degradation by XRN1.

To extend this in vitro model to physiological conditions, we evaluated the activity of NUDT2 mammalian cells. In human cells, NUDT2 localizes to both the nucleus and the cytoplasm (Supplementary Fig. 4a), which suggests that NUDT2 can be biologically active in both compartments. NUDT2 precipitated from HEK293T cells could release significantly more phosphates from a PPP-RNA compared to NUDT2 E58A mutant or control proteins (Supplementary Fig. 4b, c), indicating that mammalian cell-expressed NUDT2 also has phosphatase activity.

Targeting endogenous *NUDT2* in HeLa cells using CRISPR/Cas9 did not lead to any phenotypic alteration of cells compared to control treatments (Supplementary Fig. 6a). We transfected in vitro transcribed HCV-luciferase PPP-RNA into these cells and analyzed its abundance over time using RT-qPCR. As expected, the abundance of HCV-Luc PPP-RNA declined over time (Fig. 5c). Remarkably, in the absence of NUDT2, HCV-Luc PPP-RNA was considerably more stable, suggesting that NUDT2 might be involved in the degradation of viral RNA. To assess whether the observed effect is transferable to another cell type, we co-transfected PPP-RNA and capped-RNA into Hep3B cells or Hep3B cells lacking functional NUDT2, XRN1, or both. At 1 and 4 h post-transfection, NUDT2 depletion led to an increase in PPP-RNA stability as compared to the control (NTC) (Fig. 5d). Capped-RNA was not affected by NUDT2 depletion indicating specificity for degradation of PPP-RNA substrates. Importantly, depletion of XRN1 stabilized capped-RNA and PPP-RNA to a similar extent, underlining that XRN1 is in principle required to degrade both types of RNA. Similarly, cells co-depleted for NUDT2 and XRN1 showed increased PPP-RNA as well as capped-RNA levels. Notably, co-depletion did not result in synergistic effects compared to deleting single genes, further indicating that NUDT2 and XRN1 are operative in the same pathway (Fig. 5d). To further validate these findings, we infected Hep3B cells, depleted for NUDT2, XRN1, or NUDT2 and XRN1 with VSV-GFP (Fig. 5e). Both the lack of NUDT2 and XRN1 and the co-depletion led to a similar increase in GFP expression compared to the control, again pointing towards co-operative activities between NUDT2 and XRN1.

**NUDT2 depletion in mice accelerates the growth of PPP-RNA generating viruses in vitro.** We generated NUDT2 deficient mice from targeted embryonic stem cells (Supplementary Fig. 5a). Correct targeting of the *Nudt2* locus was confirmed by PCR (Supplementary Fig. 5b). Homozygous knockout mice were viable, bred with expected Mendelian ratios (Supplementary Fig. 5c), and did not show an obvious phenotype under

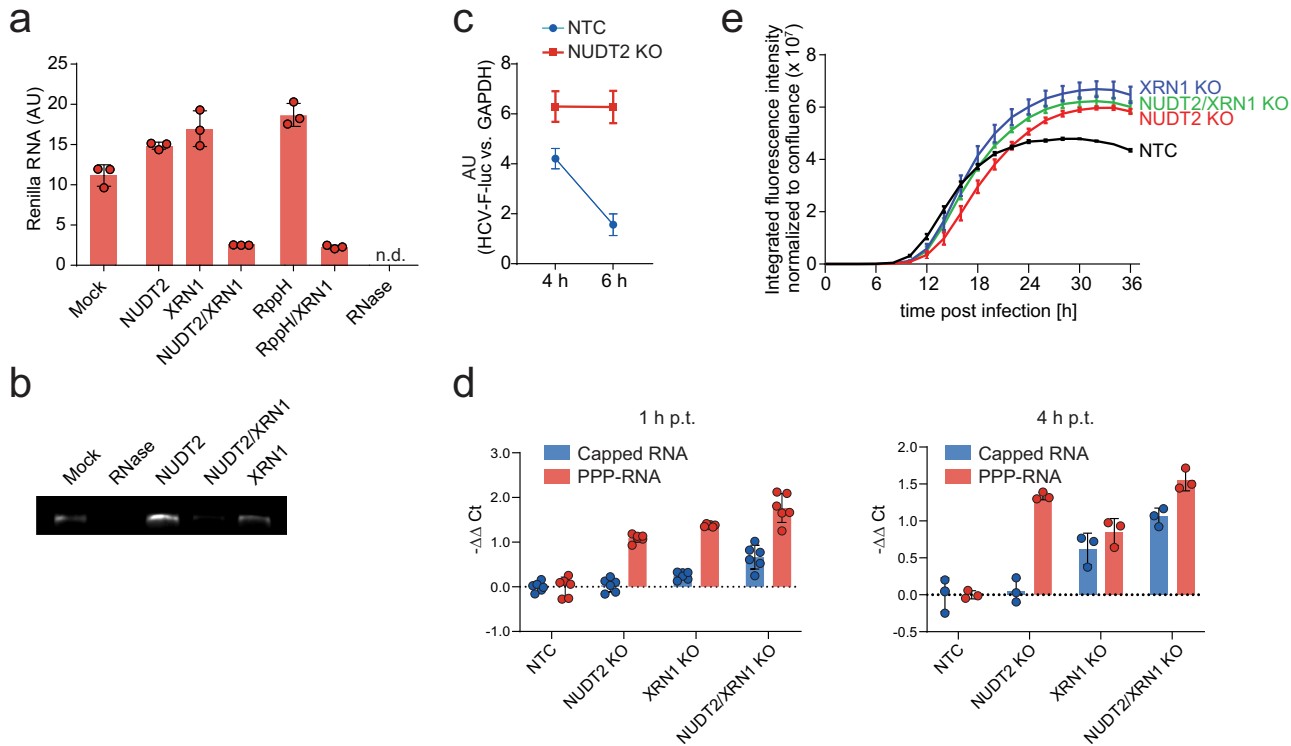

**Fig. 5 NUDT2 is preparing RNA to serve as XRN1 substrate. a** In vitro transcribed PPP-renilla RNA was left untreated (mock) or incubated with 600 nM NUDT2, 1 U XRN1, 600 nM NUDT2 together with 1 U XRN1, 5 U RppH, or 5 U RppH together with 1 U XRN1 and 10 U RNase for 4 h. The remaining RNA was quantified by RT-qPCR. The bar plot shows the fold change compared to input RNA as the mean ± SD of three technical replicates. **b** In vitro synthesized HCV RNA was treated with 5 U RNase, 600 nM NUDT2, 1 U XRN1, or co-treated with 600 nM NUDT2 and 1 U XRN1 and then analyzed on an agarose gel. **c** In vitro transcribed HCV RNA encoding luciferase was electroporated into HeLa CRISPR KO cells lacking NUDT2 or HeLa cells treated with a non-targeting-control vector (NTC). Viral RNA load was quantified 4 and 6 h post electroporation by the abundance of firefly luciferase RNA. **d** In vitro transcribed triphosphorylated renilla RNA and capped firefly luciferase RNA were co-electroporated into Hep3B CRISPR/Cas9 KO cells lacking *NUDT2*, *XRN1*, or both *NUDT2* and *XRN1*, or Hep3B cells treated with a non-targeting-control vector (NTC). RNA load was quantified 1 and 4 h post electroporation by quantifying the abundance of firefly luciferase or renilla RNA. **e** Hep3B treated with CRISPR/Cas9 lentivectors targeting *NUDT2*, *XRN1*, *NUDT2*, and *XRN1* or with a non-targeting-control vector (NTC) were infected with VSV-GFP at MOI 0.01. GFP expression was monitored using automated live-cell microscopy. The data points are displayed as the mean of five technical replicates ± SD. Shown is one representative experiment of three independent experiments.

non-infected conditions. To test whether NUDT2 deficiency may generally affect vulnerability to virus infections, we performed an unbiased proteome analysis of *Nudt2*$^{-/-}$ and *Nudt2*$^{+/+}$ bone marrow cells. This analysis allowed a parallel identification of 6288 proteins in total (Supplementary Table 2). As expected, NUDT2 could not be identified in the NUDT2 deficient mice (Fig. 6a). However, NUDT2 deficiency did not lead to significant differences in basal expression levels of the mouse proteome (considering 2866 proteins identified in 3 of 6 samples, FDR = 0.05, S0 = 0.1) (Supplementary Fig. 5d and Supplementary Table 2). Deficiency in the degradation of aberrant nucleic acids can lead to the accumulation of antiviral responses. However, compared to wild-type or heterozygous mice, *Nudt2*$^{-/-}$ mice did not show spontaneous upregulation of mRNA encoding for classic antiviral proteins (e.g., Ifit3), inflammatory cytokines (e.g., IL-6), or U1 snRNA in heart, liver, lung, or spleen (Supplementary Fig. 5f) showing that NUDT2 depletion does not affect the basal immune status of mice. NUDT2 is known to cleave Ap4A enzymatically, a dinucleotide generated under teratogenic conditions and during translational stress. To exclude that virus infection would elicit Ap4A synthesis and that accumulation of this substance could be responsible for suppressing antiviral responses in cells, we tested the influence of Ap4A on the growth of Semliki forest virus (SFV), a virus that is highly sensitive to antiviral activities of IFNs. Extracellular application of Ap4A or its

delivery into HeLa cells through lipofection impaired virus replication in a comparable manner as the double-stranded RNA analog polyI:C (Supplementary Fig. 5e), indicating that accumulation of Ap4A is not responsible for increased virus growth. Collectively, NUDT2 depletion does not result in a general perturbation of the cellular innate immune response, and increased Ap4A levels caused by NUDT2-deficiency do not promote virus growth. These observations are in line with a more direct effect of NUDT2 on viral RNA, i.e., the dephosphorylation of triphosphorylated RNA and its subsequent degradation.

We isolated total bone marrow (BM) from *Nudt2*$^{-/-}$ and littermate control mice and tested for replication of VSV wt and the IFN-stimulating variant VSV(M51A) (VSV-M2). Notably, accumulation of infectious virus particles was 10- to 30-fold increased in supernatants of bone marrow-derived from *Nudt2*$^{-/-}$ mice compared to corresponding *Nudt2*$^{+/+}$ littermate controls (Fig. 6b). Similarly, compared to controls, MEFs from *Nudt2*$^{-/-}$ animals showed up to a 100-fold increase in VSV release into the supernatant and higher accumulation of VSV-N RNA (Fig. 6c, d). The increased virus growth also correlated with an increased virus-induced cytopathic effect in *Nudt2*$^{-/-}$ compared to *Nudt2*$^{+/+}$ MEFs (Fig. 6e). Importantly, infectious virus particles and viral RNA accumulation of HSV-1 (DNA virus), which does not generate PPP-RNA, were comparable in *Nudt2*$^{-/-}$ and *Nudt2*$^{+/+}$ MEFs (Fig. 6f, g). Similarly, SFV (a virus predominantly sensed by

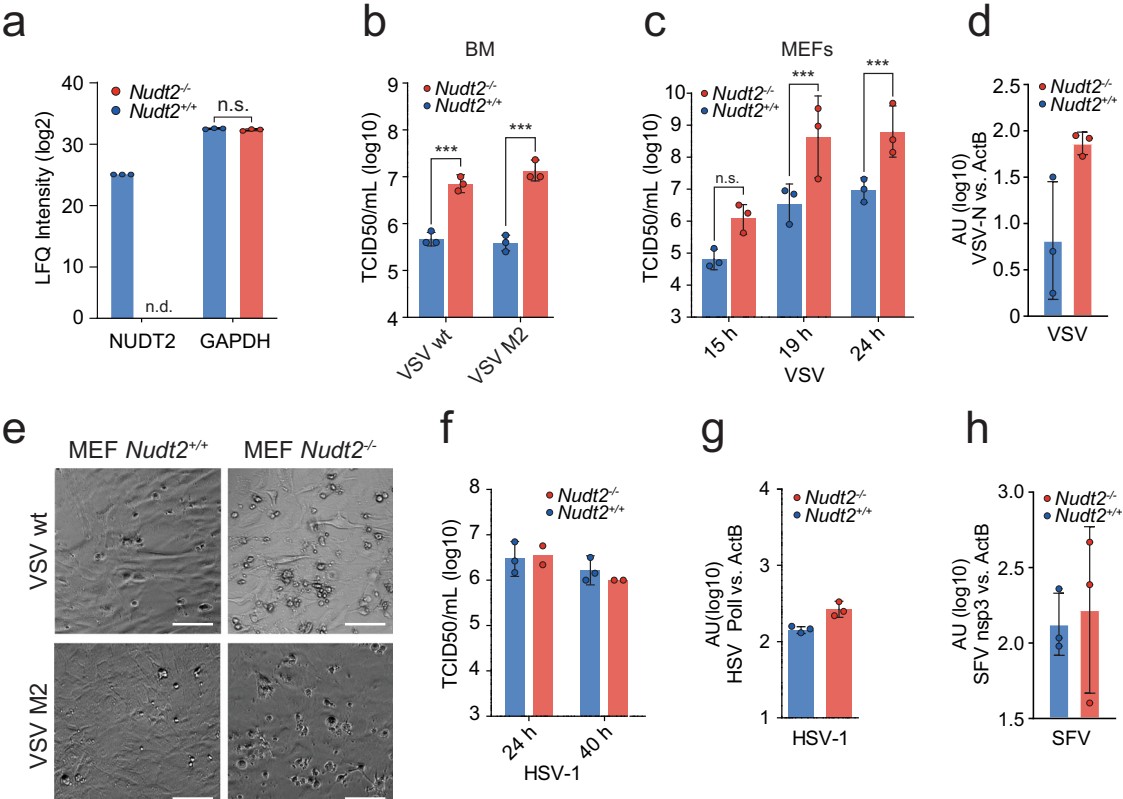

**Fig. 6 Impact of NUDT2 on virus growth. a** Bone marrow cells were isolated from *Nudt2* knockout (*Nudt2⁻/⁻*), and control mice were tested for the abundance of NUDT2 by mass spectrometry. The mean LFQ intensities ± SD for NUDT2 and GAPDH of six biological replicates are shown. A two-sided students t-test (FDR 0.05) was performed. **b** Accumulation of infectious virus particles in supernatants of *Nudt2+/+* or *Nudt2⁻/⁻* bone marrow infected with either wild-type VSV, VSV-M2 at an MOI of 1 for 16 h. The average TCID50/mL of three biological replicates ± SD are shown. *** $p < 0.001$ two-way ANOVA with Bonferroni's post-test. **c** Accumulation of VSV in supernatants of *Nudt2+/+* and *Nudt2⁻/⁻* MEFs infected with VSV at an MOI of 0.001 for 16 h. Average TCID50/mL of three biological replicates ± SD. *** $p < 0.001$, two-way ANOVA with Bonferroni's post-test. **d** Viral RNA load in *Nudt2+/+* or *Nudt2⁻/⁻* MEFs infected with VSV. RNA levels were quantified by RT-qPCR analysis using specific primers for the VSV nucleoprotein transcript. Data were normalized to murine *ActB* RNA. The mean ± SD of three biological repetitions are shown in arbitrary units. **e** Phase-contrast micrographs of *Nudt2+/+* or *Nudt2⁻/⁻* MEFs infected with VSV for 16 h. The scale bars represent 100 μm. MEF cells of the indicated genotype were infected with HSV-1 (**f**, **g**) or SFV (**h**) at an MOI of 2 for 16 h. **f** Accumulation of infectious virus particles was analyzed. The graph shows the average TCID50/mL of three biological replicates ± SD. **g**, **h** Accumulation of viral RNA quantified by RT-qPCR analysis using specific primers for the DNA polymerase I of HSV or nsp3-transcript of SFV. Data were normalized to murine *ActB* RNA. Bar plots show average arbitrary units of three biological repeats ± SD.

MDA5)[23] replicated equally well in *Nudt2⁻/⁻* compared to *Nudt2+/+* MEFs (Fig. 6h). We concluded from these studies that mouse NUDT2 impairs the growth of PPP-RNA generating viruses in vitro, but it does not affect the accumulation of other viruses.

We proceeded to infect *Nudt2+/+* and *Nudt2⁻/⁻* mice with VSV using an intranasal infection regime, causing encephalitis. Surprisingly, despite significant growth differences in cell culture experiments, the infection in vivo did not lead to significant differences in survival rates when comparing *Nudt2+/+* and *Nudt2⁻/⁻* mice (Supplementary Fig. 6a). Moreover, the viral RNA load in the cerebellum of these mice was not significantly different. (Supplementary Fig. 6b) as well as transcripts for ISGs like Ifit3 or cytokines like IL-6 (Supplementary Fig. 5f). These experiments suggest that additional proteins with redundant functions are operative on an organismal level.

## Discussion

RNA turnover is a highly conserved process and tightly controlled to regulate RNA abundance and to target erroneous RNAs for degradation. PPP-RNA, as present in viral genomes and to some extent in bacteria[10], should, in principle, be protected from canonical cellular 5′-3′ mRNA decay pathways present in higher eukaryotes. However, the half-life of PPP-RNA delivered into cells is only ~0.5–1.0 h, which is relatively short compared to the 9 h average half-life of cellular mRNA[6,24]. Although the PPP-group protects PPP-RNA from XRN1-dependent degradation, depletion of this exonuclease increases viral RNA half-life[25]. This suggested the presence of a triphosphatase that converts PPP-RNA into an XRN1 substrate[4,26,27]. Indeed, recently the phosphatases DUSP11 and DOM3Z (DXO) were identified as proteins that convert PPP-RNA into P-RNA, which serves as a substrate for XRN1[8,9,28]. Since RNA degradation is an evolutionary, highly conserved process, we asked whether proteins known to be involved in PPP-RNA degradation in bacteria are also functionally active in mammalian cells. Using a siRNA screen, phosphate release assays, automated live-cell microscopy, and RT-qPCR, we identified that the protein NUDT2 could release phosphates from 5′-PPP RNA substrates and contributes to control the degradation of viral PPP-RNA. We show that NUDT2 is active on a broad range of PPP-RNA substrates but has a preference for processing longer substrates. NUDT2 sequentially releases two phosphates, and a monophosphorylated RNA is generated that can then serve as a substrate for the cellular exonuclease XRN1. This suggests that NUDT2 could be active in the degradation process of RNAs derived from diverse viruses.

The Nudix hydrolase RppH has been shown to play a role in the dephosphorylation of 5′-PPP and mRNA degradation RNA in bacteria[11]. Although PPP-mRNA is present in bacteria, recent reports of Luciano et al. suggest that *E. coli* predominantly contain diphosphorylated mRNA, which appears to be the preferred substrate of RppH in *E. coli*[10]. The high structural similarity of RppH and mammalian NUDT2 indicates similar activities of both proteins, which is supported by the ability of NUDT2 and RppH to process phosphorylated RNA. Likely, NUDT2 has a preference for longer RNA constructs (compare Figs. 1b–e, 3a with Figs. 3c, 4a–c) and thus can also act on viral dipho-sphorylated RNA, characteristic for some viruses, including reoviruses[29], supported by the consecutive PPP-RNA depho-sphorylation mechanism employed by NUDT2. Notably, NUDT2 is conserved throughout vertebrates, flies, worms, and bacteria, suggesting a broad functionality of this protein: Depending on the organism NUDT2 can be involved in physiological RNA turnover as well as antiviral defense mechanisms. NUDT2 deficiency in haploid KBM7 cells has been proposed to elicit an increased ISG expression pattern, which should result in reduced virus growth[30]. However, this reported phenotype does not match our observations in various diploid human and murine cells, the phenotype of Nudt2 knockout mouse, or the viral growth phe-notype in Nudt2 depleted systems.

The importance of RNA degradation is highlighted by the fact that many viruses evolved mechanisms to escape RNA decay pathways and potentially hide from NUDT2 activity. Potential evasion strategies include capping of RNA by either employing the virus-derived capping machinery or seizing it from cellular mRNAs in a process called cap snatching[31,32]. Other RNA viru-ses, like viruses of the family Caliciviridae, protect their RNA by attaching a viral genome-linked protein (VPg). RNA of some viruses like Picornaviruses carry a structural element named internal ribosome entry site (IRES) that could also present a barrier for exonucleases like XRN1[33]. HCV employs an uncon-ventional mechanism to protect its RNA from degradation as it recruits the liver-specific microRNA miR-122 to its 5′-end. Through complementary binding to the HCV RNA, miR-122 protects the RNA from degradation by XRN1[6,34] by sterically hindering XRN1 binding to the 5′-end. Although NUDT2 dephosphorylated all tested RNA substrates, it may be that the activity of NUDT2 is similarly inhibited by steric hindrance under very special conditions, as seen for XRN-I. Flaviviruses form membranous replication factories that cover viral RNA from cellular proteins[35], which may also be an evasion strategy to escape NUDT2 activity. It is not yet clear if NUDT2 can access such intracellular sub-compartments or if the activity of NUDT2 is limited to free 5′-PPP RNAs in the cytoplasm. Screening the impact of NUDT2 on more viruses would greatly enhance our understanding of the impact of NUDT2 dependent viral restric-tion and RNA decay. Since we could not observe a significant difference in survival rates between VSV-infected *Nudt2*$^{+/+}$ and *Nudt2*$^{-/-}$ mice, we cannot exclude that other cellular pyropho-sphatases have redundant functions. Such redundancies have also been reported for the activity of pattern recognition receptors (PRRs) in vivo. Despite the requirement of certain PRRs to induce cytokines and reduce virus growth in vitro, in vivo phe-notypes were often less convincing. The RNA sensor Toll-like receptor (TLR) 7, for instance, only reveals its in vivo phenotype in the absence of the RIG-I sensing pathway[36]. Similarly, the in vivo activity of TLR3 strongly depends on the infection route of the pathogen[37]. Moreover, in an infection model for the DNA virus mouse Cytomegalovirus, the DNA sensor TLR9 only shows an effect in livers but not in spleens, clearly indicating an organ-specific effect[38]. Similarly shown for PRRs, the in vivo phenotype of NUDT2 may only be visible when using sophisticated in vivo

models or when specific infection models are used. On an orga-nismal level, organ-specific expression differences or viral counter mechanisms could affect the impact of NUDT2.

Antiviral responses are often governed by proteins that are regulated by type-I IFNs. The expression of NUDT2 protein, however, does not change after cytokine treatment or virus infection. This may suggest that NUDT2, in addition to its ability to dephosphorylate viral RNA, may also have additional house-keeping functions. Interestingly, RNA Polymerase III products, including 5S ribosomal RNA, non-coding, and in some cases coding RNAs, should contain 5′-PPP termini. It is conceivable that proteins, such as NUDT2, DOM3Z (DXO), and DUSP11, are processing such aberrant RNAs or other RNA molecules to avoid stimulation of the innate immune system in the absence of pathogen encounter. Since self-RNA recognition may have dra-matic effects on the cell, NUDT2, DUSP11, and DOM3Z (DXO) may fulfill redundant functions under steady-state conditions. Indeed, we could not obtain cells that lack both NUDT2 and DUSP11, despite the successful depletion of either of the two proteins. The full activity of these proteins may be required in case of viral infections since the abundance of viral PPP-RNA may exceed the catalytic capacity of either of these proteins.

## Methods

**Cells, reagents, and viruses**. HeLa (ATCC CCL-2) and HEK293T (ATCC CRL-3216) cells were described previously and were authenticated by ATCC[39]. Hep3B cells were obtained from Prof. R. Bartenschlager, University Hospital Heidelberg (Germany). Mouse embryonic fibroblasts (MEFs) were isolated from 13.5-day old embryos from heterozygous breeding pairs. Murine bone marrow was isolated from mouse femur, and tibia and cultured in DMEM supplemented with 10% fetal calf serum (FCS, GE Healthcare), 10 mM HEPES pH 7.4, 1 mM sodium pyruvate, and 2 mM L-glutamine. Cell lines were maintained in DMEM (PAA Laboratories) containing 10% FCS and antibiotics (100 U/mL penicillin, 100 μg/mL streptomy-cin). All viruses used are classified as BSL2 pathogens in Germany, and all experiments were carried out according to official regulations. Influenza A/SC35M-NS1-2A-renilla-2A-NEP (IAV-ren)[40], VSV, VSV-M2[41], VSV-GFP[42], VSV-AV3-GFP[19] have been previously described. HSV-1 (F-strain) was from Soren Riis Paludan (Uni Aarhus, Denmark), VSV-Luc was a gift from Gert Zimmer (Uni-versity Bern, Switzerland), and SFV was a gift from Andres Merits (University of Tartu, Estonia). Recombinant XRN1, CIP, and RppH and restriction enzymes for in vitro transcriptions were obtained from NEB, single nucleotides for in vitro transcriptions and phosphate release assays were from Jena Bioscience, RNasin was from Promega, TURBO DNase was obtained from Thermo Fisher Scientific, and T7 RNA polymerase was produced by the Core Facility of the MPI of Biochemistry. The CellTiter-Glo assay kit and the luciferase substrate d-Luciferin (E1602) were purchased from Promega. siRNAs were purchased from Qiagen or were synthe-sized by the Core Facility of the MPI of Biochemistry (see Supplementary Table 1).

**Preparation of triphosphorylated or capped RNA**. As RNA substrate, IVT4, a defined double-stranded triphosphorylated RIG-I ligand, was used as described[43]. To test the sequence specificity of NUDT2, RNA substrates with varying starting nucleotides, including single-stranded and double-stranded 5′-ends, were used as described[13]. In vitro transcribed RNA was generated from linear, double-stranded DNA templates, hybridized of complementary oligodeoxynucleotides (Supple-mentary Table 3). Except, the template for the generation of renilla RNA (Promega, E2231), which PCR amplified according to the manufacturer's instructions (2x HF Mastermix, NEB M0541S) with the corresponding oligodeoxynucleotides (Sup-plementary Table 3). The linear dsDNA product was purified (Monarch PCR Cleanup, T1030S) and directly used for in vitro transcription. RNA Sequences beginning with 5′-AG contained a T7 φ2.5 promoter; all other templates contained a T7 φ6.5 promoter. The antisense strand of the DNA template was annealed with sense-oriented oligos covering the respective T7 promoter region[44]. These oligo-nucleotides were mixed 1:1 to a final concentration of 10 μM each in 10 mM Tris (pH 7.5–8.0), 50 mM NaCl, and 1 mM EDTA, heated to 95 °C for 5 min and cooled down to 20 °C at a rate of 0.1 °C/s on a PCR cycler. For the preparation of renilla luciferase RNA, the pRL-SV40 plasmid was linearized by BamHI cleavage. To prepare firefly luciferase RNA, the RiboMAX Large Scale RNA Production Systems Kit (Promega) control plasmid was used. Transcription was carried out overnight at 37 °C according to the manufacturer's protocol of the T7 RiboMAX Large Scale RNA Production Systems kit (Promega). After DNase treatment for 30 min at 37 °C using 1 U of RNase-free DNase per μg of DNA template, ivtRNA was purified using either the MinElute Cleanup kit (Qiagen) or the Monarch RNA Cleanup Kit (NEB).

For the preparation of HCV PPP-RNA, the pFK-JFH plasmid DNA[45] was linearized by MluI cleavage. Transcription was carried out for 6–8 h at 37 °C in

5× transcription buffer (200 mM Tris pH 8, 30 mM MgCl$_2$, 10 mM Spermidine, and 5 mM DTT), 2.5 mM ATP, 2.5 mM UTP, 2.5 mM CTP, 2.5 mM GTP, RNasin (Promega), and 1000 U T7 RNA polymerase (Core Facility of the MPI of Biochemistry), in 100 µL final volume. 2 µL TURBO DNase (Thermo Fisher Scientific) was added for 30 min at 37 °C. Reactions were purified with mini Quick Spin RNA Columns (Roche) according to the manufacturer's protocol. For the 44-mer single-stranded RNA, the pGEX-6P-T7-44-mer-RNA[46] plasmid linearized by EcoRI cleavage served as a template. RNA was synthesized by in vitro transcription according to the MEGAscript T7 Kit (Thermo Fisher Scientific). After incubation for 16 h at 37 °C and TURBO DNase (Thermo Fisher Scientific) treatment for 30 min at 37 °C, the RNA was purified according to the Direct-zol RNA extraction kit (Zymo Research). For the generation of double-stranded RNA, a 20-nt antisense RNA (Metabion) (5′-acucucucucucucuccc-3′) was annealed to the in vitro-transcribed RNA. In vitro transcribed RNA was capped using the ScriptCap Cap 1 Capping System (CellScript, C-SCCS1710).

**siRNA-mediated knockdown experiments**. Duplex siRNAs were transfected using either siPrime transfection reagent (GE Healthcare) for the Nudix hydrolase siRNA screen or Metafectine Pro transfection reagent with the SI$^+$Buffer (Biontex) for targeted gene knockdowns. Transfection was performed according to the manufacturer's instructions for HeLa cells. Briefly, we transfected 15 pmol of pooled siRNAs per 1e5 cells in a 24 well format. Cells were infected 48 h after siRNA knockdown. Duplex siRNAs were either purchased from Qiagen or synthesized by the Core Facility at the MPI of Biochemistry (Supplementary Table 1). Cell viability after siRNA-mediated knockdown was assessed with the CellTiter-Glo (Promega) luminescent cell viability assay, according to the manufacturer's protocol.

**Virus infections and determination of virus titers and growth**. To determine the impact of *NUDT2* knockout on virus growth, MEFs and bone marrow were seeded the day before infection. Cells were infected with VSV wt at a multiplicity of infection (MOI) of 0.001, SFV at an MOI of 0.001, HSV-1 at an MOI of 2, VSV-Luc at an MOI of 0.001, IAV-renilla at an MOI of 0.1, and HSV-Luc at an MOI of 2. Bone marrow was infected with VSV wt and VSV M2 at an MOI of 1. Twenty-four hours post-infection or as indicated specifically in the figure legend, cells were harvested for quantitative RT-qPCR, Western Blot or and the supernatant was harvested to test for virus accumulation by 50% tissue culture-infective dose (TCID50) assays on Vero E6 cells. To quantify the effects of siRNA-mediated knockdown of Nudix hydrolases, cells were infected 48 h after the knockdown was performed with VSV-Luc at an MOI of 0.001 for 24 h, IAV-renilla at an MOI of 1 for 48 h, VSV-GFP at an MOI of 0.001 for 24 h, and VSV-M2-GFP at an MOI of 0.0001 for 24 h. GFP levels were determined fluorometrically. Z-scores were calculated from three independent experiments individually for every gene and virus, as previously described[47].

For RT-qPCR analysis, RNA was isolated using the NucleoSpin RNA Plus kit (Macherey Nagel) according to the manufacturer's protocol. For proteomic analysis, cell pellets were snap-frozen in liquid nitrogen before further processing. For western blot analysis, cells were lysed in Laemmli buffer, boiled for 10 min at 95 °C, and subjected to SDS-polyacrylamide gel electrophoresis and western blot analysis. For GFP-tagged viruses, virus growth was measured by determining GFP levels in a microplate reader (Tecan Infinite 200 Pro) with an excitation wavelength of 485 nm and an emission wavelength of 535 nm. For luciferase-tagged viruses, cells were lysed in Passive Lysis Buffer (Promega), and virus load was determined by measuring firefly or renilla-luciferase. VSV-GFP growth kinetics were monitored and analyzed using the IncuCyte S3 Live-cell imaging system and software (Essen Bioscience). Heb3b cells, transduced with a non-targeting control vector or with a CRISPR/Cas9-mediated depletion of either *NUDT2*, *XRN1*, or *NUDT2* and *XRN1*, were infected with VSV-GFP at an MOI of 0.01 in five technical replicates. For each technical replicate, two phase-contrast and green fluorescence images were acquired by a 10× objective. Fluorescence images were further processed to remove the background signal using a top-hat filter. The integrated intensity of GFP+ cells per image was then normalized to the total area of cells by division for each time point.

**Protein expression and purification**. DNA sequences of *NUDT2*, *NUDT2 E58A*, *NUDT12*, *NUDT12 E369A*, *NUDT14*, *NUDT14 E121A*, *NUDT17*, *NUDT17 E124A* were ordered as gBlocks from IDT containing overhangs suitable for SLIC cloning. All constructs were cloned into pCoofy4, providing an N-terminal His6-MBP tag. Protein expression and purification were performed by the Core Facility of the MPI of Biochemistry. Briefly, plasmids were transformed in BL21-AI pRARE bacteria, and protein expression was carried out in an autoinduction medium containing 0.2% arabinose. Cells were lysed using an AVESTIN high-pressure homogenizer, and cleared lysate was used for protein purification using a HisTrap HP column (GE Healthcare: 17-5247-01) and further purified by gel filtration using a Superose 6 PC 3.2/30 (GE Healthcare: 17-0673-01) (mobile phase: 20 mM Tris-HCl pH 7.5, 250 mM NaCl, 10 mM MgCl$_2$). Proteins were dialyzed with D-Tube Dialyser Midi (cutoff 6–8 kDa) in a buffer containing 20 mM Tris-HCl pH 7.5, 250 mM NaCl, 10 mM MgCl$_2$, and 1 mM DTT. The identity of recombinant proteins was confirmed by mass spectrometry.

To test phosphate release from mammalian expressed NUDT2 cells, N-terminally 6-myc-tagged NUDT2, NUDT2 E58A, and NS1 of Influenza A virus (with mutated RNA binding domain (R38A, K41A) of strain A/PR8/34) were transiently expressed from pCS2-N-6xMyc plasmids. For this, HEK293T cells were transfected using 1.3 µg of plasmid per 1e6 cells and a ratio of the transfection reagent PEI to plasmid DNA of 3:1. The cells were collected after two days, washed 3× with PBS, pelleted, and frozen. Subsequently, the cells were lysed with TAP lysis buffer (50 mM Tris-HCl at pH 7.5, 5% glycerol, 0.2% NP-40, 1.5 mM MgCl2, 100 mM NaCl, supplemented with complete protease and phosphatase inhibitor cocktails (Roche) on ice for 30 min, followed by sonication (10 min, 4 °C) and clearing of the lysate (4000 g, 30 min, 4 °C). The expressed, Myc-tagged proteins were then precipitated using 80 µL slurry of equilibrated α-myc agarose beads (Chromotek, yta-20). Beads were washed three times with NEB 2 reaction buffer to equilibrate for the phosphate release assay.

**SDS-PAGE and western blotting**. Validation of precipitation of myc-tagged proteins: Immunoprecipitated proteins were eluted in the presence of 1x SDS sample buffer (62.5 mM Tris-HCl, 2% SDS, 10% glycerol, 50 mM DTT, 0.01% bromophenol blue, pH 6.8) and by boiling the samples at 95 °C for 5 min. Eluted proteins were then separated on a 4–12% Bis-Tris acrylamide gel (NuPAGE, Invitrogen) at 100 V for 90 min and transferred onto a 0.2 µm nitrocellulose membrane by wet-blotting at 300 mA for 1 h. The membrane was then blocked with 5% milk, 3× washed with PBS with 0.25% Tween-20 (PBS-T) for 5 min each, and stained with primary antibodies in 5% milk followed by incubation with an appropriate secondary antibody.

Primary antibodies used in this study were as follows: monoclonal mouse α-NUDT2 (Santa Cruz: sc-271410, 1:1000 dilution), monoclonal mouse α-β-Actin-HRP (Santa Cruz: sc-47778, 1:1000 dilution), monoclonal mouse α-myc-HRP (Roche, 11814150001, 1:2000 dilution), and secondary horseradish peroxidase (HRP)-coupled antibody rabbit α-mouse IgG (Cell Signaling, 7076, 1:2000 dilution).

**Phosphate release assay**. For time-resolved analysis, the phosphatase activities of Nudix hydrolases, CIP, or RppH were analyzed with the EnzChek Phosphate Assay (E6646, Sigma-Aldrich) according to the manufacturer's instructions. For optimal performance, the following adjustments were made: one replicate (200 µL reaction volume) included 600 nM NUDT protein or the corresponding mutant, 4.5 µg of RNA substrate, 0.5 U of PNP, 0.2 mM MESG, and NEB 2.0 buffer. This setting was validated to yield optimal assay performance (Supplementary Fig. 3a). The absorbance was measured at 360 nm every three min. The phosphate release was calculated from a linear standard curve specific for each time point.

For single time-points, the ability of Nudix hydrolases to release phosphates from RNA was tested with in vitro transcribed RNA (ivtRNA), and ribonucleotide triphosphates (rNTPs). The malachite green assay was performed according to the manufacturer's instructions (MAK307, Sigma-Aldrich) after incubating 900 ng IVT4 with NUDT2 or NUDT2 E58A in NEB2 buffer for 3 h at 37 °C. This setting was validated to yield optimal assay performance. In order to test NUDT2 activity on a panel of possible substrates, NUDT2 (600 nM) and NUDT2 E58A (600 nM) were incubated for 3 h with rATP, rGTP, rUTP, and rCTP (2.5 µM) for 3 h at 37 °C in NEB2 reaction buffer. CIP (5 U) (NEB) was used as a positive control for released inorganic phosphate in all experiments. For immunoprecipitated NUDT2, NUDT2 E58A or NS1 (IAV) derived from HEK cell lysate, 20 µL of beads were incubated in 40 µL NEB 2 reaction buffer, including 900 ng IVT4, for 3 h at 37 °C. Beads were separated (500 g, 1 min, RT), and the phosphate release assay was performed using the malachite green assay as described above.

**Phosphate release from γ-$^{32}$P-labeled RNA**. γ-$^{32}$P labeled RNA was synthesized by in vitro transcription according to the MEGAscript T7 Kit (Thermo Fisher Scientific) supplemented with 0.17 µM [γ-$^{32}$P] GTP (Perkin Elmer). For the 44-mer single-stranded RNA, the pGEX-6P-T7-44mer-RNA[46] plasmid linearized by EcoRI cleavage served as a template. For the dsRNA IVT4, two annealed primer pairs (Supplementary Table 3)[43] were used as a template. After incubation for 16 h at 37 °C and DNase digestion with TURBO DNase (Thermo Fisher Scientific) for 30 min at 37 °C, the RNA was purified according to the Direct-zol RNA extraction kit (Zymo Research). For the generation of a double-stranded RNA of the 44-mer ssRNA, a 20-nt antisense RNA (Metabion) (5′-acucucucucucucuccc-3′) was annealed to the in vitro transcribed RNA.

The labeled RNAs were incubated with purified NUDT2 (600 nM), NUDT12 (600 nM), NUDT14 (600 nM), NUDT17 (600 nM), and RppH (5 U) purchased from NEB in a solution (20 µL) containing NEB2 buffer at 37 °C. Reaction samples (10 µL) were taken at 0 and 90 min and quenched with 2.5 µL of EDTA (100 mM, pH 8.0), analyzed by TLC on PEI-cellulose (Merck Millipore), and developed with potassium phosphate buffer (0.3 M, pH 7.5). Spot intensities were imaged by using a GE Typhoon FLA 9000 Imager. For time-course experiments, the labeled RNA was incubated with purified NUDT2 or NUDT2 E58A (600 nM) in a solution (50 µL) containing NEB2 Buffer for 0–90 min at 37 °C. Reaction samples (10 µL) were quenched at time intervals with 2.5 µL of EDTA (100 mM, pH 8.0) and analyzed by TLC on PEI-cellulose (Merck Millipore) developed with potassium

phosphate buffer (0.3 M, pH 7.5). Spot intensities were imaged by using a GE Typhoon FLA 9000 Imager.

**Synthesis of pppApG and pppGpA.** Dinucleotides were synthesized using DMT-2′-O-TBDMS-rA(bz), DMT-2′-O-TBMDS-rG(ib) amidites, and rA(Bz) and rG(ibu) CPG (Chemgenes) on a 10 μM scale by using standard solid-phase oligoribonucleotide coupling techniques. The CPG-bound dimers were triphosphorylated using the cyclotriphosphate protocol of triphosphate synthesis[48], except that after deprotection of the dinucleotides, the reaction mixture was converted into the triethylammonium salt using Dowex Et3NH+ before reverse-phase chromatography on a Source RP column. DecNHpppApG and DecNHpppGpA RP-HPLC product peaks were converted into pppApG and pppGpA as described. The pure pppGpA and pppApG were isolated as triethylammonium salts by ion-exchange chromatography on a Hi Screen DEAE FF column using a gradient of 0.0–0.6 M Triethylammonium bicarbonate. The integrity of the pppApG and pppGpA dimers was verified by LC-MS analysis.

**HPLC analysis.** A short RNA (pppGpA, 0.1 OD260) was incubated for 2 h at 37 °C in a solution (15 μL) containing NEB2 buffer (1×) with recombinant NUDT2 (600 nM), NUDT2 E58A (600 nM), RppH (5 U) or left untreated. Reactions mixtures were quenched with 3.75 μL of EDTA (100 mM, pH 8.0) and diluted 1:20 with water. For time-course measurements, the two short RNAs (pppGpA or pppApG, 0.1 OD260 each) were incubated with purified recombinant NUDT2 (600 nM), NUDT14 (600 nM), and NUDT2 E58A (600 nM), respectively, in a solution (75 μL) containing NEB2 buffer (1×) at 37 °C. Reaction samples (15 μL) were taken at 0, 1, 3, 6, and 19 h, quenched with 3.75 μL of EDTA (100 mM, pH 8.0), and diluted 1:20 with water. Aliquots (80 μL) were analyzed by reversed-phase (RP)-HPLC on an AGILENT 1100 System using an XBridge C18 column (3.5 μm, 150 × 2.1 mm, WATERS). Chromatography was performed using a two-eluent buffer system. Buffer A consists of an aqueous solution of 100 mM Et3NHOAc, pH 7.8, and buffer B consists of an aqueous solution of 100 mM Et3NHOAc, pH 7.8, and 40% (v/v) MeCN. Chromatograms were recorded at a wavelength of 260 nm. HPLC was performed using the following gradient condition: 0% B over 4 min, to 15% B over 20 min, to 18% B over 16 min to 100% B over 10 min, held at 100% B for 15 min, to 0% B over 2 min, held at 0% B for 13 min with a flow rate of 0.3 mL/min. Quantification of eluted compounds was performed by integration of the corresponding peak areas.

**Analysis of RNA integrity and transcript level analysis.** To assess RNA integrity after treatment with recombinant Nudix hydrolases, small reaction samples were analyzed by the Bioanalyzer (Agilent) using the Agilent Small RNA kit according to the manufacturer's protocol. For quantitative RT-qPCR analysis, total RNA from cells was isolated using the NucleoSpin RNA Plus kit (Machery-Nagel). RNA from mouse cerebellum, heart, liver, lung, or spleen was extracted using Lysing Matrix M and the FastPrep-24 (MP Biomedicals) instrument. Organs were directly lysed in LBP buffer, and RNA isolation was continued using the NucleoSpin RNA Plus kit (Machery-Nagel). Next, 200–500 ng of RNA was reverse-transcribed with Prime-Script RT Master Mix (TAKARA) and after that quantified by RT-qPCR using the QuantiFast SYBR Green RT-PCR Kit (Qiagen) and a CFX96 Touch Real-Time PCR Detection System (Bio-Rad). Each cycle included 10 s at 95 °C and 30 s at 60 °C, followed by a melting curve analysis. Primer sequences are depicted in Supplementary Table 4.

**Generation of CRISPR/Cas9 knockout cells.** Double-stranded 30-nt guide sequences (Supplementary Table 5) targeting the human *NUDT2* gene and a non-targeting control were designed and cloned into pLentiCRISPRv2 (Addgene #52961) as previously described[49,50]. HEK 293T cells were co-transfected with the obtained plentiCRISPR v2 vector (6 μg), pxPAX2 (3 μg), and pMD2G (1.5 μg) in a 10 cm dish using PEI. 24 h post-transfection, the medium was replaced with 10 mL of fresh DMEM. Seventy-two hours post-transfection, the medium was collected, filtered through a 0.45 μm pore size membrane (Millipore), and HeLa cells were infected with the lentivirus. Twenty-four hours post-infection, the medium was replaced with DMEM containing puromycin (1 μg/mL) for four passages. Knockout efficiency was analyzed by western blotting.

**RNA degradation assays.** Degradation of 1000 ng of an in vitro transcribed renilla or HCV PPP-RNA carrying a firefly luciferase was assessed by incubating it either with 10 U RNase A, 600 nM recombinant NUDT2, 1 U recombinant XRN1, 5 U recombinant RppH, or both 600 nM NUDT2 and 1 U XRN1 or 5 U RppH and 1 U XRN1 for 4 h at 37 °C. To determine RNA levels after incubation, renilla RNA was quantified using RT-qPCR, and the presence of HCV PPP-RNA was assessed using 1% agarose gels. The stability of in vitro transcribed HCV PPP-RNA encoding luciferase was determined by electroporation into HeLa CRISPR/Cas9 KO cells lacking *NUDT2* or HeLa cells treated with a non-targeting control vector. HCV PPP-RNA load was quantified 4 and 6 h post electroporation by quantifying the abundance of firefly luciferase RNA by RT-qPCR. To determine degradation of capped or PPP-RNA, capped firefly luciferase and PPP-renilla luciferase were mixed in equimolar amounts and were co-electroporated into Hep3B CRISPR/Cas9 KO cells lacking *NUDT2*, *XRN1*, or both *NUDT2* and *XRN1*, or Hep3B cells

treated with a non-targeting-control vector. Cells were incubated for 1 and 4 h at 37 °C. To determine RNA levels after incubation, renilla or firefly luciferase RNA was quantified using RT-qPCR and normalized to RPLP0 RNA.

**Ap4A treatment.** HeLa cells were transfected with 1 μg poly(I:C) using 1 μL METAFECTENE Pro or 250 nmol Ap4A using 2 μL METAFECTENE Pro or 250 nmol of Ap4A was added directly to the media[51,52]. As controls, the cells were stimulated with PBS in combination with and without METAFECTENE Pro. Twenty-four hours post-treatment, the cells were infected with SFV6-2SG Nano-Luc at an MOI of 0.0035 or left uninfected for 24 h. The cells were lysed in Passive Lysis Buffer, and virus growth was assessed by measurement of renilla luciferase.

**Generation of *Nudt2* knockout mice.** The *Nudt2* allele was rendered unfunctional by introducing a poly(A) signal after exon 1 in a construct purchased from the European Conditional Mouse Mutagenesis consortium (EUCOMM). This genetically modified ES cell clone (Clone ID: EPD0146_2_H06, Cell type: JM8.N4) was injected into C57BL/6 blastocyst donors. Male chimeras were bred with C57BL/6 females to produce heterozygous *Nudt2*+/tm1a mice (*Nudt2*+/−) containing a tm1a cassette encoding for the lacZ gene, a neomycin resistance, a splice acceptor, and a polyA site flanked by FRT sites. Heterozygous animals were then bred to generate homozygous *Nudt2*tm1a/tm1a mice (*Nudt2*−/−).

For genotyping NUDT2 knockout mice, ear punches were collected from 21 to 23-days old mice and genotyped by PCR using the following primers: Nudt2_fwd: 5′-ccagctttcttgtacaaagtgg-3′, Nudt2 wt_fwd: 5′-gaatttctgctgctgcaggc-3′, Nudt2_rev: 5′-cctagtgaagggacaaagcagc-3′ and using the following parameters: 94 °C for 1 min, followed by 35 cycles of 94 °C for 30 s, 65 °C for 20 s, and 72 °C for 1 min. The wild-type allele was detected as a band at 492 bp, whereas the inserted tm1a cassette was detected as bands of 417 and 912 bp.

**Ethics statement and mouse infections.** All animal experiments were performed in compliance with the German Animal Welfare Law (TierSchG BGBl. S. 1105; 25.05.1998). The mice were housed and handled in accordance with good animal practice as defined by FELASA. All animal experiments were approved by the responsible state office (Landesverwaltungsamt Sachsen-Anhalt), the University of Magdeburg, under permit number AZ 42502-2-1344. During intranasal infections (i.n), 8–12-week-old mice were first anesthetized by i.p. injection with a mixture of ketamine (100 mg/g body weight) and xylazine (5 mg/g body weight). They then were infected with 5e6 plaque-forming units (PFU) of VSV in 20 μL PBS unless otherwise indicated. Mice that lost more than 20% of their body weight were sacrificed.

**Proteomic analysis of *Nudt2* knockout in mouse bone marrow cells.** For proteomic analysis, wild-type and *Nudt2* knockout bone marrow cells (5e6/cell line/replicate; 4 replicates) were lysed in 300 μL of lysis buffer (4% SDS, 10 mM DTT in 50 mM Tris pH 7.6) supplemented with complete protease inhibitor (Roche) and boiled for 10 min at 95 °C. The protein concentration of cleared cell lysates was assessed using Pierce 660 nm Protein Assay (Thermo Fisher Scientific) according to the manufacturer's protocol. Protein concentrations for all cell lines and replicates were equalized to 35 μg protein content and alkylated with 55 mM IAA for 20 min in the dark. Proteins were precipitated with acetone and resuspended in 150 μL of denaturation buffer (6 M Urea, 2 M Thiourea in 10 mM HEPES pH 8.0). Protein digestion was performed by adding 1:100 (protein:enzyme) trypsin and LysC overnight at room temperature. Peptides were purified on stage tips and analyzed by LC-MS/MS using the EASY-nLC 1200 system coupled to a Q Exactive HF mass spectrometer (Thermo Fisher Scientific). Peptide mixtures were separated on a 50 cm C18-reversed-phase column (Reprosil-Pur 120 C18-AQ, 1.9 μM, 200 × 0.075 mm; Dr. Maisch) using a 180-minute linear gradient of 5−30% buffer B (0.1% formic acid and 80% ACN) with a flow rate of 250 nL/min. The mass spectrometer was set up to run a Top10 method acquiring full scans (300–1,600 m/z, R = 60,000 at 200 m/z) at a target of 3e6 ions, followed by isolation of the ten most abundant ions, HCD fragmentation (target 1e5 ions, maximum injection time 120 ms, isolation window 1.4 m/z, NCE 27%, and underfill ratio of 20%) and detection in the Orbitrap analyzer.

**Raw data processing and statistical analysis.** Peptide identification and quantification were performed using MaxQuant (version 1.6.17.0). For all MaxQuant searches, typical default parameters were employed. Spectra were searched against forward and reverse sequences of the reviewed mouse proteome, including isoforms (Uniprot, UP000000589). Settings included carbamidomethylation of cysteine as fixed modification and oxidation of methionine and N-terminal protein acetylation as variable modifications. Trypsin/P was specified as the proteolytic enzyme. The minimal peptide length was defined as 7 amino acids, and match-between-run was disabled for LFQ determination.

The protein groups were further analyzed with Perseus (1.6.14.0). The matrix included 6288 proteins and was filtered with the default settings: potential contaminant only identified by site and reverse. After log2 transformation, a two-sided students *t*-test (FDR 0.05) was performed, including proteins with at least three valid values (in total) and imputed (width 0.3 and downshift 1.8).

**Phylogenetic analysis**. To establish the phylogenetic relationship between different human Nudix hydrolases and bacterial RppH, protein sequences from all human Nudix hydrolases and *E. coli* RppH were analyzed using MAFFT[53] with standard settings. The resulting tree was displayed using Dendroscope[54] and visually adapted using Adobe Illustrator CS6 (16.0.3).

**Multiple sequence alignment**. Protein sequences of NUDT2 from *Homo sapiens*, *Danio rerio*, *Caenorhabditis elegans*, *Mus musculus*, *Xenopus laevis*, *Gallus gallus*, and *Drosophila melanogaster* were collected and aligned using the Clustal Omega tool with standard settings. The multiple sequence alignment was visualized using ESPript 3.0[55].

**Statistical analysis**. All data were analyzed either using GraphPad Prism (8.4.3) or R (4.0.2) using R Studio (1.3.1056). Data acquired with the Incucyte live-cell imaging platform were analyzed using IncuCyte Analysis Software (2019B Rev2).

**Reporting summary**. Further information on research design is available in the Nature Research Reporting Summary linked to this article.

## Data availability

The data supporting the findings of this study are available from the corresponding authors upon reasonable request. Source data for the figures and supplementary figures are provided as a Source Data file. The LC-MS/MS data and MaxQuant output generated in this study have been deposited in the ProteomeXchange Consortium via the PRIDE partner repository[56] under accession code P9. Source data are provided with this paper.

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

## Acknowledgements
We thank the core facility of the MPI of biochemistry for support. We further thank Soren Riis Paludan for HSV-1, Gerd Zimmer for VSV-Luc and Fabien Bonneau for advice on the EnzChek assay. Work in the authors' laboratories was supported by an ERC consolidator grant (ERC-CoG ProDAP, 817798), the Federal German Ministry for Education and Research (BMBF; COVINET), the Bavarian State Ministry of Science and Arts (Bavarian Research Network FOR-COVID), and the German Research Foundation (PI 1084/3, PI 1084/4, PI 1084/5 and TRR179/TP10, TRR237/A07) to A.P. Work of P.S. at the Leibniz Institute for Experimental Virology (HPI) was supported by the Free and Hanseatic City of Hamburg and the Federal Ministry of Health. The research of M.B. and D.W. was supported by the German Research Foundation (TRR179).

## Author contributions
B.T.L., K.K., Q.E., L.L.A., A.R., C.D., K.M., A.K., S.M., and P.S. conducted the experiments. B.T.L., K.K., Q.E., L.L.A., and A.R. analyzed the data. J.L., D.W., M.M., and M.B. contributed critical reagents. B.T.L., K.K., Q.E., and A.P. designed the experiments and wrote the paper.

## Funding

## Competing interests
The authors declare no competing interests.
