## [Peer Review File · Nature Communications]

Title: NUDT2 initiates viral RNA degradation by removal of 5'-phosphatesREVIEWER COMMENTS

Reviewer #1 (Remarks to the Author):

Laudenbach et al report on the NUDT2 protein. They describe how this protein is capable of removing triphosphate from the 5' end of RNAs, in vitro and in cell lines. They argue that the reason viruses conserve triphosphates are that those stabilizes viral RNA, by protecting them from XRN1 mediated degradation. They show that the deletion of the NUDT2 gene improves viral growth in vitro, but are unable to duplicate those result in a mouse KO model. Despite the lack of a clear in vivo phenotype, they do present a convincing data set in vitro that I believe merit publication in Nature Com. However; there are some aspect of the biochemical work that needs clarification.

Figure 1c, it is difficult for the reader (at least this one) to follow the data-processing that lead to this figure, therefore it is also difficult to follow the conclusion, could the author please explain in more details how this figure was generated, in particular how the Z score was calculated.

The concentration of PPP-nucleotides used in figure 3d are rather low (2.5 micro molar) compared to the physiological concentration of ATP. Do the author either have a convincing argument for the low concentration used or could they repeat the exp at physiological relevant concentrations. The author should also compare the concentration used here to figure 4e.

In figure 4e, the authors argue that they look at release of PPi but this is not true, they look at the remaining nucleotide after degradation with NUDT2 and from that they try to infer mechanism. The text needs to more clearly reflect this. They also need to explain how they decided to use dinucleotide after showing that single nucleotide was not a NUDT2 substrate, do those dinucleotide properly act like a true RNA substrate? They also at some occasion refer to the dinucleotide as an RNA, I would prefer that they stick with dinucleotide.

Reviewer #2 (Remarks to the Author):

In virus infected host cells, viral RNA degradation is critical to prevent virus replication and to protect host cell survival. Mammalian cells contain a various mechanisms to detect and degrade endogenous or exogenous RNAs including the Nudix motif containing proteins. This manuscript focuses on the Nudt2 nudix protein which was first reported to asymmetrically hydrolyze Ap4A to yield AMP and ATP and then reported to be m7G-cap decapping protein. The authors show Nudt2 functions in the decay of uncapped RNA with a 5' triphosphate to remove the first two phosphates and generate 5' end monophosphated RNA that can be degraded by the XRN1 nuclease. Since many viral RNAs are not capped, they propose this activity functions on invading viral RNAs and show Nudt2 phosphate affected RNA virus growth in host cells. This is a very interesting observation and could be important against viral defense. However, a shortcoming is the amount and more significantly, the time frame of the in vitro experiments which are

carried out for several hours. The long time frame of the assays raises significant concerns with the interpretation of the data whether they results are truly due to Nudt2 or minor copurifying contaminants that become evident after 3-4 hour incubations.

1. It is difficult to interpret the phosphatase assays in figure 2. Depending on the reaction volume, it appears 500-1000ng of recombinant protein is used in the assays with 3 hours incubation time. Both of these (especially the time) raise question about the derived conclusions. Error bars are not presented (which should be) but nevertheless, the mutant having half the level of activity as the wild type protein with so much protein and time raises concerns whether the observed activity is due to a minor copurifying contaminant in the preparations (possibly even contamination by RppH). For example, data from wild type and mutant Nudt17 is not particularly different than that of wild type Nudt2. The reaction conditions should be optimized to enable activity detection with lower protein concentrations and substantially less time for the reactions. Ideally, the experiments should be repeated with a time course, rather than a single time to more accurately deduce the pyrophosphatase activity of each protein.

2. Figure 3: Similar concerns for the amount of protein and duration of the assays applies to this figure as outline above in point 1.

3. Figure 5a: Again, similar concerns for the amount of protein and duration of the assays applies to this figure as outline above in point 1 (here 4 hr incubation is used). There is a high likelihood that these results are due to minor copurifying contaminants rather than the Nudix protein themselves.

4. Figure 5c: In this figure the authors transfect in vitro transcribed luciferase RNA into Nudt2 knockout or WT cells and follow the decay of the transfected RNA at 4 and 6 hours by qRT-PCR. I don't believe a conclusion can be made from this figure. As detailed in numerous occasions in the literature (see Wang, Cell 2001 for one of the first), long RNAs lacking a protective group on the 3' end have half-lives measured in minutes, not hours. Any RNA remaining at 4 and 6 hours is likely residual background detected by the sensitivity of the RT-PCR assay. To be interpretable, these experiments will need to be repeated with shorter time frame. In addition, how are the transfection efficiencies normalized in these assays and how are the values compared to each other?

Similar concern is also true for figure 5d which detects transfected RNA after 4 hours. This is especially true since Nudt2 has previously been reported to have m7G cap decapping activity in vitro.

5. Figures 2 and 3: Error bars and statistical significance should be presented for all the data figures.

6. Nudt5, Nudt21 and Nudt22 knockdown caused a negative impact on VSV growth shown in Fig 1c. The authors explained this phenotype could be caused by a reduced cell viability. However, the cell viability assay shown in Supplementary Figure 1, knockdown Nudt5, Nudt21 or Nudt22 does not influence cell viability. Please clarify.

7. In figure 1d and 1e, the authors state that NUDT12, -14 and NUDT17 exhibited inconsistent results on growth of FluAV virus in HeLa and A549 cells. Although the knockdown efficiency is provided for these mRNAs in Supplementary Figure 1b for HeLa cells, the knockdown efficiencies in A549 cells is not shown. This is important since statistically significant increase is observed in the Nudt12 knockdown in HeLa cells, but not A549 cells.

8. It is difficult to extrapolate data obtained by the pppGpA dinucleotide to that of a longer RNA with a triphosphate 5' end. Many of the Nudix proteins function on dinucleotides, but only few function on similar structures on the 5' end of an RNA (see Song et al., 2013, RNA). This distinction should explicitly be made since any activity observed on pppGpA may not exist on a RNA with 5' triphosphate.

9. Page 19, line 450: Despite the initial report in Celesnik (2008, Nature) that RppH removes PPi from a 5' triphosphate, it was subsequently shown to be a phosphatase that primarily removes a monophosphate from diphosphorylated 5' end RNA (see Luciano et al. 2017, Mol.Cell). Perhaps the lack of ppGA product in figure 4e is due to the single time point used in the assay. Similar to Nudt2, sequential removal of the phosphates might be detected with a shorter time course.

10. Sup. Fig 2b: it is difficult to conclude Nudt2 is nuclear from the images presented since these do not appear to be confocal images.

11. Use of the DOM3Z terminology was confusing. The official HGNC name of this protein is Dxo.

Reviewer #3 (Remarks to the Author):

Deprotection of the 5' end is often a key step in RNA degradation. In this manuscript, Laudenbach et al. report that the mammalian Nudix hydrolase NUDT2 is able to convert 5'-terminal RNA triphosphates to monophosphates that render the RNA vulnerable to 5'-to-3' exonucleolytic degradation. They also investigate the substrate specificity and reaction products of this enzyme. NUDT2 deficiency engineered in cell lines, bone marrow cells, and mouse embryonic fibroblasts is shown to increase the growth of RNA viruses whose genomes and transcripts bear a 5' triphosphate, but it has little effect on the pathogenicity of these viruses in mice, perhaps due to functional redundancy with other RNA pyrophosphatases such as DUSP11 and/or DOM3Z.

This manuscript addresses an interesting topic of potential significance for understanding host defenses against some important RNA viruses. However, the evidence as to the specificity of NUDT2 is not yet persuasive.

Major issues:

1. The experimental results in Fig. 2b, 2d, 3a, 3b, and 3d are described as representative examples of

three independent experiments. However, by not graphing the mean and standard deviation of all three measurements, the authors have made it impossible for readers to judge the statistical significance of any observed differences.

2. Fig. 2b, 3a, 3b. It is surprising that wild-type NUDT2 and the NUDT2-E58A mutant appear to be so similar in reactivity, differing by only 2-4 fold despite the loss of a key active site glutamate. This suggests either of two possibilities. One is that the purified enzymes were contaminated by a phosphatase of some kind, although the poor reactivity of NUDT2 with mononucleotides (Fig. 3d) and of the mutant enzyme with single-stranded RNA and pppGpA (Fig. 4c, 4h) argues otherwise. A more likely explanation is that the fixed-time assays used to compare the reactivity of the various substrates in Fig. 3a and 3b were allowed to proceed too far to completion, obscuring differences in substrate reactivity. This conclusion is supported by the negligible increase in phosphate release when the enzyme concentration was doubled. The reactivity of these substrates should instead be compared by monitoring the progress of each reaction as a function of time. Time-course assays of this kind would likely reveal greater differences in reactivity and provide more conclusive evidence as to the substrate specificity of NUDT2.

Additional comments:

3. Line 36, 77. RppH is an abbreviation of RNA pyrophosphohydrolase, not RNA pyrophosphatase H.

4. Line 51. Add "In eukaryotes" at the beginning of the sentence about RNA degradation by XRN1 and the RNA exosome.

5. Line 78. RNase E is a 5'-monophosphate stimulated endonuclease, not a 5'-to-3' exonuclease.

6. Line 78. In some bacterial species, RppH prepares RNA for 5'-to-3' exonucleolytic degradation by RNase J.

7. Line 374. Change "deplete mammalian Nudix genes" to "deplete mammalian Nudix proteins".

8. Fig. 2c, 3c. What is the 4-nt band in these gels? Tracking dye?

9. Lines 396, 407, 421, 436, 444, 445, 803, 806, 820, 824, 829. The single-stranded, double-stranded, and hairpin (IVT4) RNAs used as substrates should be illustrated in a supplementary figure that shows the complete sequence of each.

10. Line 407. Change "double-stranded RNA" to "single-stranded RNA", which is the substrate examined in Fig. 2d.

11. Line 424. Change "with ssRNA overhangs" to "with or without ssRNA overhangs".

12. Lines 427-428 and 459-460. The authors conclude that “the identity of the first nucleotide is not relevant for NUDT2 catalytic activity” and that “the identity of the first nucleotide does not affect NUDT2 dephosphorylation”, even though only A and G were compared at that position. These sentences should be modified to reflect the fact that only purines were tested.

13. Fig. 3a, 3b. The sequences of the RNA substrates diagrammed in these panels should be provided in a supplementary figure.

14. Fig. 3d. This graph gives the impression that RppH reacts well with double-stranded 5' ends when, on the contrary, it has previously been shown to react much more poorly with base-paired 5' ends than with unpaired 5' ends.

15. Fig. 4a, 4b, 4c, 4d. What is the basis for concluding that the high-mobility spot near the top of the TLC plates is phosphate (Pi) and not pyrophosphate? The detection of diphosphorylated RNA products in Fig. 4e, 4f, 4g, 4i, and 4j is consistent with the release of phosphate. However, the main product of the reaction of RppH with triphosphorylated RNA was previously shown to be pyrophosphate, yet in Fig. 4d the product of that reaction is roughly aligned with the calibration mark for phosphate (Pi). Is it possible that the conditions used for TLC were unable to separate phosphate from pyrophosphate? Marker lanes containing radioactive phosphate and pyrophosphate should be included in the TLC experiments to clear up this uncertainty.

16. Lines 567-568. The statement that “NUDT2 is highly conserved throughout evolution” should be toned down, as Supplementary Fig. 2b compares its sequence only in worms, flies, and vertebrates.

17. Lines 570-571. Change “mechanisms to RNA decay pathways” to “mechanisms to evade RNA decay pathways”.

18. Lines 579-580. The mechanism by which miR-122 binding would protect the 5' end of HCV RNA from attack by NUDT2 requires more explanation in light of the authors' conclusion that this enzyme reacts well with base-paired 5' ends.

19. Line 587. The acronym PRR needs to be defined.

20. Line 844, Fig. 5a. Although the legend states that “one of three biological experiments” is shown, what actually is shown are the results of triplicate experiments in vitro with purified enzymes.

General:

We want to thank the reviewers and the Editor for their feedback, which we addressed in the updated version of the manuscript. We feel that these adjustments further improved the manuscript.

Reviewer #1:

Laudenbach et al report on the NUDT2 protein. They describe how this protein is capable of removing triphosphate from the 5' end of RNAs, in vitro and in cell lines. They argue that the reason viruses conserve triphosphates are that those stabilizes viral RNA, by protecting them from XRN1 mediated degradation. They show that the deletion of the NUDT2 gene improves viral growth in vitro, but are unable to duplicate those result in a mouse KO model. Despite the lack of a clear in vivo phenotype, they do present a convincing data set in vitro that I believe merit publication in Nature Com. However; there are some aspect of the biochemical work that needs clarification.

We thank reviewer 1 for the positive comments. We amended the manuscript according to the suggestions and feel that it very much improved thanks to these comments.

- Figure 1c, it is difficult for the reader (at least this one) to follow the data-processing that lead to this figure, therefore it is also difficult to follow the conclusion, could the author please explain in more details how this figure was generated, in particular how the Z score was calculated.*

We thank reviewer 1 for pointing our attention towards this point. We now explain the Z-scoring better in figure legends and overall simplified the display. The rationale to choose the four Nudix proteins for follow up experiments was to pick the two NUDTs that mostly affected either of the two viruses used. This simplified explanation helps to understand the rationale to select the four NUDTs for follow up experiments.

- The concentration of PPP-nucleotides used in figure 3d are rather low (2.5 micro molar) compared to the physiological concentration of ATP. Do the author either have a convincing argument for the low concentration used or could they repeat the exp at physiological relevant concentrations. The author should also compare the concentration used here to figure 4e.*

We are aware that the concentration of nucleotides was below physiological levels. However, to compare NUDT2's dephosphorylation efficacy on RNA and nucleotides, we needed to use equimolar amounts of the substrates. We doubled the amount of

single nucleotides as compared to the RNA substrate but still did not get any phosphate release. We feel that this shows that the NUDT2 proteins prefer RNAs compared to nucleotides and worded it like this in the text.

- *In figure 4e, the authors argue that they look at release of PPi but this is not true, they look at the remaining nucleotide after degradation with NUDT2 and from that they try to infer mechanism. The text needs to more clearly reflect this. They also need to explain how they decided to use dinucleotide after showing that single nucleotide was not a NUDT2 substrate, do those dinucleotides properly act like a true RNA substrate? They also at some occasion refer to the dinucleotide as an RNA, I would prefer that they stick with dinucleotide.*

We thank reviewer 1 for the thoughtful comments. We now took care to change our wording and to more precisely describe the results shown in the respective figure. In Fig. 4 we needed to use dinucleotides in order to be able to discriminate tri- from di- and monophosphorylated substrates. Longer nucleotides would not have resolved on the HPLC system. However, we acknowledge in the text that longer substrates may be converted better. That dinucleotides could serve as substrates may be supported by crystal structure evidence that shows NUDT2 association to the dinucleotide AP4A, which is another well-known substrate.

Reviewer #2:

In virus infected host cells, viral RNA degradation is critical to prevent virus replication and to protect host cell survival. Mammalian cells contain a various mechanisms to detect and degrade endogenous or exogenous RNAs including the Nudix motif containing proteins. This manuscript focuses on the Nudt2 nudix protein which was first reported to asymmetrically hydrolyze Ap4A to yield AMP and ATP and then reported to be m7G-cap decapping protein. The authors show Nudt2 functions in the decay of uncapped RNA with a 5' triphosphate to remove the first two phosphates and generate 5' end monophosphated RNA that can be degraded by the XRN1 nuclease. Since many viral RNAs are not capped, they propose this activity functions on invading viral RNAs and show Nudt2 phosphate affected RNA virus growth in host cells. This is a very interesting observation and could be important against viral defense. However, a shortcoming is the amount and more significantly, the time frame of the in vitro experiments which are carried out for several hours. The long time frame of the assays raises significant concerns with the interpretation of the data whether they results are truly due to Nudt2 or minor copurifying contaminants that become evident after 3–4 hour incubations.

We thank reviewer 2 for the comments and now added a substantial amount of data to the manuscript to address the concerns regarding time-resolved RNA degradation and potential contamination issues. In addition, we provide figures for reviewer 2 that further support the validity of our data (reviewer Figure 1). We hope that reviewer 2 will agree with the conclusions we draw from these experiments.

- *1. It is difficult to interpret the phosphatase assays in figure 2. Depending on the reaction volume, it appears 500–1000ng of recombinant protein is used in the assays with 3 hours incubation time. Both of these (especially the time) raise question about the derived conclusions. Error bars are not presented (which should be) but nevertheless, the mutant having half the level of activity as the wild type protein with so much protein and time raises concerns whether the observed activity is due to a minor copurifying contaminant in the preparations (possibly even contamination by RppH). For example, data from wild type and mutant Nudt17 is not particularly different than that of wild type Nudt2. The reaction conditions should be optimized to enable activity detection with lower protein concentrations and substantially less time for the reactions. Ideally, the experiments should be repeated with a time course, rather than a single time to more accurately deduce the pyrophosphatase activity of each protein.*

Reviewer 2 is concerned regarding the phosphate release assay presented in figure 2. We agree that the previous version of the manuscript did not contain optimization data that we performed in order to obtain the best possible results. We now add in supplementary figure 1c additional data that shows NUDT2 incubation at two different concentrations for three different time points. We can clearly see that the assay does not give reliable results after 1h and is not saturated after 3h of NUDT2

incubation with RNA. We, therefore, opted to use a 3h incubation of RNA with NUDT2 for optimal performance of this assay.

Finally, we now added “total mass” proteomics data of the purified proteins, which indicate high purity of the recombinant proteins isolated from bacteria (Supplementary Figure 1d). We are certain that the observed phosphatase activity is not coming from RppH since this potential “contaminant” should lead to a PPP-RNA > P-RNA conversion by the release of 2 phosphates (Fig 4e), which we do not observe for the NUDT2 protein. Also, recombinant NUDT2 does contain RNases that could potentially release phosphates since RNA does not get degraded by recombinant NUDT2, even after long incubation time (Fig. 2c, 3b, 5a, b).

However, to exclude any potential doubts on the quality of the recombinant protein, we purified NUDT2 from HEK293 cells, and we used this alternative enrichment for phosphate release assays. Importantly, we could demonstrate the same phosphate release as we could observe for the recombinant proteins. We believe that this experiment proves that the activity of bacterial NUDT2 is not associated with unwanted co-purified bacterial products but a real function of the protein itself.

In addition, we ask reviewer 2 to consider figure 4, which shows gamma phosphate release in a very different assay with a shorter time frame.

Previous publications (Ge, Honghua, 2013) had used the E58A mutation, which made us introduce the same mutation in NUDT2. Similar to the results presented previously, the E58A mutant also had some rest-activity, likely due to other amino acids in the catalytic center that are sufficient to orient the substrate towards the active site.

We added error bars and indicated biological repeats throughout the manuscript – we thank reviewer 2 to point this out.

- *2. Figure 3: Similar concerns for the amount of protein and duration of the assays applies to this figure as outline above in point 1.*

Please see the answer to point 1. Since we found a similar activity for HEK293 and E.coli derived NUDT2, we are certain about the protein quality. Moreover, we further optimized the assay and got highly reproducible results.

- *3. Figure 5a: Again, similar concerns for the amount of protein and duration of the assays applies to this figure as outline above in point 1 (here 4 hr incubation is used). There is a high likelihood that these results are due to minor copurifying contaminants rather than the Nudix protein themselves.*

We would like to refer to the above answer. In addition, here, we show that only the combination of the two proteins leads to the degradation of RNA, which was what we predicted. The single proteins did not have an effect. In our opinion, it would be

highly unlikely that this complex experimental setup could be explained by the combination of contaminants (which would have to potentiate their activity).

- *4. Figure 5c: In this figure the authors transfect in vitro transcribed luciferase RNA into Nudt2 knockout or WT cells and follow the decay of the transfected RNA at 4 and 6 hours by qRT-PCR. I don't believe a conclusion can be made from this figure. As detailed in numerous occasions in the literature (see Wang, Cell 2001 for one of the first), long RNAs lacking a protective group on the 3' end have half-lives measured in minutes, not hours. Any RNA remaining at 4 and 6 hours is likely residual background detected by the sensitivity of the RT-PCR assay. To be interpretable, these experiments will need to be repeated with shorter time frame. In addition, how are the transfection efficiencies normalized in these assays and how are the values compared to each other?*

We took this concern very seriously. During the revision process, we extensively tested the degradation of transfected RNA in our experimental system. We show here as a figure to reviewer 2 that we can detect luciferase RNA decay even after 6 hours after transfection (Reviewer figure 1a, b). Furthermore, we provide data by Binder and Sulaimanov et al. (2013, PLoS Pathog.), indicating a half-life time of a replication-deficient HCV genome of 0.92 hrs (Reviewer Figure 1c). Reviewer 2 is absolutely correct that there is substantial decay of the transfected RNA during this time. Since we are interested in the degradation of RNA, the late time points allow us to particularly study the stability of the RNA in the absence of NUDT2 or XRN-1. In addition, Fig 5d shows additional decay of the transfected RNA between 4 to 6h, suggesting that the transfected RNA was still present and subsequently degraded.

- *Similar concern is also true for figure 5d which detects transfected RNA after 4 hours. This is especially true since Nudt2 has previously been reported to have m7G cap decapping activity in vitro.*

Similar answer as for Fig. 5c.

- *5. Figures 2 and 3: Error bars and statistical significance should be presented for all the data figures.*

We added all error bars to all experiments and statistical analysis for relevant comparisons.

- *6. Nudt5, Nudt21 and Nudt22 knockdown caused a negative impact on VSV growth shown in Fig 1c. The authors explained this phenotype could be caused by a reduced cell viability. However, the cell viability assay shown in Supplementary Figure 1, knockdown Nudt5, Nudt21 or Nudt22 does not influence cell viability. Please clarify.*

In the previous version of the manuscript, we said that some of the phenotypes may in part be explained with differential cell viability. This referred to reduced cell viability after NUDT21 and particularly NUDT22 depletion. We now adjusted the text so that it is less ambiguous. It now reads: “Cell viability, was tested by cell titer glow assay (Supplementary Fig. 1a). NUDT5, NUDT19, NUDT21 and NUDT22 showed a negative impact on VSV growth (Fig. 1c). For NUDT19 and NUDT22 this phenotype could potentially be explained by reduced cell viability after depletion of these NUDTs.”. We want to thank reviewer 2 for this comment. We feel that the new text explains the data better.

- *7. In figure 1d and 1e, the authors state that NUDT12, -14 and NUDT17 exhibited inconsistent results on growth of FluAV virus in HeLa and A549 cells. Although the knockdown efficiency is provided for these mRNAs in Supplementary Figure 1b for HeLa cells, the knockdown efficiencies in A549 cells is not shown. This is important since statistically significant increase is observed in the Nudt12 knockdown in HeLa cells, but not A549 cells.*

This is correct. In the revised version of the manuscript, we removed the A549 data since it did not add valuable information. In follow-up experiments, we were focussing on NUDT2, 12, 14, 17, and the A549 cell data was not used for any further selection of NUDTs.

- *8. It is difficult to extrapolate data obtained by the pppGpA dinucleotide to that of a longer RNA with a triphosphate 5' end. Many of the Nudix proteins function on dinucleotides, but only few function on similar structures on the 5' end of an RNA (see Song et al., 2013, RNA). This distinction should explicitly be made since any activity observed on pppGpA may not exist on a RNA with 5' triphosphate.*

This is a valid point. In the updated version of the manuscript, we are more careful with extrapolating the dinucleotide dephosphorylation experiments to longer RNAs. We also discuss that the data obtained by dinucleotide dephosphorylation needs to be confirmed on larger RNAs, in the best case on viral RNA.

- *9. Page 19, line 450: Despite the initial report in Celesnik (2008, Nature) that RppH removes PPi from a 5' triphosphate, it was subsequently shown to be a phosphatase that primarily removes a monophosphate from diphosphorylated 5' end RNA (see Luciano et al. 2017, Mol.Cell). Perhaps the lack of ppGA product in figure 4e is due to the single time point used in the assay. Similar to Nudt2, sequential removal of the phosphates might be detected with a shorter time course.*

We thank the reviewer for these important references but are not sure whether we understand this point correctly. For NUDT2, we do see the removal of single phosphates due to the detection of diphosphorylated nucleotides (Fig. 4e). Also, the

multiple hour time-resolved analysis (Fig. 4f, g) suggests that NUDT2 facilitates a sequential phosphate removal from dinucleotides *in vitro*. Concerning RppH, Luciano et al. described the RppH-dependent dephosphorylation mechanism in bacteria. Their data imply that RppH cannot remove a monophosphate from triphosphorylated RNA transcripts but performs a rather slow one-step conversion from triphosphorylated RNA to monophosphorylated RNA by the release of a pyrophosphate. This could be an explanation of why we do not observe a diphosphorylated intermediate as in Fig. 4e when treating with RppH. In this manuscript, we focus on NUDT2, for which we can clearly demonstrate monophosphate release (Fig. 4e). We now explain the differences between RppH and NUDT2 in the updated version of the manuscript in a more precise way.

- *10. Sup. Fig 2b: it is difficult to conclude Nudt2 is nuclear from the images presented since these do not appear to be confocal images.*

We thank the reviewer for this comment and corrected the figure legend. The images are confocal images.

- *11. Use of the DOM3Z terminology was confusing. The official HGNC name of this protein is Dxo.*

We thank the reviewer for pointing this out. We changed this accordingly.

Reviewer #3:

Deprotection of the 5' end is often a key step in RNA degradation. In this manuscript, Laudenbach et al. report that the mammalian Nudix hydrolase NUDT2 is able to convert 5'-terminal RNA triphosphates to monophosphates that render the RNA vulnerable to 5' to 3' exonucleolytic degradation. They also investigate the substrate specificity and reaction products of this enzyme. NUDT2 deficiency engineered in cell lines, bone marrow cells, and mouse embryonic fibroblasts is shown to increase the growth of RNA viruses whose genomes and transcripts bear a 5' triphosphate, but it has little effect on the pathogenicity of these viruses in mice, perhaps due to functional redundancy with other RNA pyrophosphatases such as DUSP11 and/or DOM3Z.

This manuscript addresses an interesting topic of potential significance for understanding host defenses against some important RNA viruses. However, the evidence as to the specificity of NUDT2 is not yet persuasive.

Major issues:

- *1. The experimental results in Fig. 2b, 2d, 3a, 3b, and 3d are described as representative examples of three independent experiments. However, by not graphing the mean and standard deviation of all three measurements, the authors have made it impossible for readers to judge the statistical significance of any observed differences.*

We thank Reviewer 3 for this important comment. We replaced all affected figures accordingly and included at least 3 replicates. The standard deviation and appropriate statistical analysis are shown.

- *2. Fig. 2b, 3a, 3b. It is surprising that wild-type NUDT2 and the NUDT2-E58A mutant appear to be so similar in reactivity, differing by only 2-4 fold despite the loss of a key active site glutamate. This suggests either of two possibilities. One is that the purified enzymes were contaminated by a phosphatase of some kind, although the poor reactivity of NUDT2 with mononucleotides (Fig. 3d) and of the mutant enzyme with single-stranded RNA and pppGpA (Fig. 4c, 4h) argues otherwise. A more likely explanation is that the fixed-time assays used to compare the reactivity of the various substrates in Fig. 3a and 3b were allowed to proceed too far to completion, obscuring differences in substrate reactivity. This conclusion is supported by the negligible increase in phosphate release when the enzyme concentration was doubled. The reactivity of these substrates should instead be compared by monitoring the progress of each reaction as a function of time. Time-course assays of this kind would likely reveal greater differences in reactivity and provide more conclusive evidence as to the substrate specificity of NUDT2.*

Ge, Honghua, (2013) had used the E58A mutation, which made us introduce the same mutation in NUDT2. In their hands, the E58A mutant also had some rest-activity,

likely due to other amino acids in the catalytic center that are sufficient to orient the substrate towards the active site. We, therefore, believe that additional mutations would be required to totally abrogate NUDT2 activity. We now provide additional evidence for the protein quality, which suggests high purity of the recombinant proteins (Supplementary Fig. 1d) and could not detect contaminating phosphatases (Fig. 4e-g). To eliminate any concern of co-purification issues, we isolated NUDT2 and NUDT2 E58A from HEK293T cells and reproduced the same phosphate release (Fig. 2f). While the negative controls did not release substantial amounts of phosphates, isolated NUDT2 released phosphates from PPP-RNA. Since we used this additional system that relies on purification from a eukaryotic cell line rather than bacteria, it is unlikely that the observed effects are due to contaminating phosphatases.

To demonstrate that our fixed time point is efficient for the determination of overall reactivity, we now provide an additional time-course experiment in Sup. Fig. 1c, which shows phosphate release at 1, 3, and 6h of incubation. While the early time point (e.g., 1h) did not result in robust amounts of released phosphates, we got solid signals at 3h of incubation. At this time point, the saturation was not reached since additional phosphates were released between 3-6h. We think that the dynamic range is sufficient to detect major differences in substrate specificity. Since the phosphate release assay requires substantial amounts of the substrate, we could not perform similar kinetics for the individual RNAs for all the individual substrates used in Fig 3a. If there were substantial differences in substrate specificity, we would be able to detect them in the fixed term assay. This is nicely exemplified in Fig 3c, which shows no release of phosphates from single nucleotides. Reviewer 3 is right that kinetic assays would be important if we were trying to ask whether NUDT2 preferential reactivity on a given substrate. We feel that the exact substrate specificity and reactivity would best be studied in future work that also considers additional features such as naturally occurring secondary structures or nucleotide modifications.

Additional comments:

- 3. Line 36, 77. *RppH* is an abbreviation of RNA pyrophosphohydrolase, not RNA pyrophosphatase *H*.

The names and inconsistencies have been updated.

- 4. Line 51. Add “*In eukaryotes*” at the beginning of the sentence about RNA degradation by *XRN1* and the RNA exosome.

We added “in eukaryotes” as suggested.

- 5. Line 78. RNase E is a 5' -monophosphate stimulated endonuclease, not a 5' -to-3' exonuclease.

Thank you, this was changed accordingly.

- 6. Line 78. In some bacterial species, RppH prepares RNA for 5' -to-3' exonucleolytic degradation by RNase J.

We now added RNase J to the scheme and the text.

- 7. Line 374. Change “deplete mammalian Nudix genes” to “deplete mammalian Nudix proteins”.

This is well taken, we changed accordingly.

- 8. Fig. 2c, 3c. What is the 4-nt band in these gels? Tracking dye?

Yes, it is an internal control used for aligning and normalization between the different samples.

- 9. Lines 396, 407, 421, 436, 444, 445, 803, 806, 820, 824, 829. The single-stranded, double-stranded, and hairpin (IVT4) RNAs used as substrates should be illustrated in a supplementary figure that shows the complete sequence of each.

The sequence of these substrates had been provided in Supplementary Table 3.

- 10. Line 407. Change “double-stranded RNA” to “single-stranded RNA”, which is the substrate examined in Fig. 2d.

We replaced this figure. The new figure 2d uses single-stranded RNA, which is now clearly stated in the figure legends.

- 11. Line 424. Change “with ssRNA overhangs” to “with or without ssRNA overhangs”.

Updated.

- 12. Lines 427-428 and 459-460. The authors conclude that “the identity of the first nucleotide is not relevant for NUDT2 catalytic activity” and that “the identity of the first nucleotide does not affect NUDT2 dephosphorylation”, even though only A and G were compared at that position. These sentences should be modified to reflect the fact that only purines were tested.

Thank you very much. This was updated accordingly.

- *13. Fig. 3a, 3b. The sequences of the RNA substrates diagrammed in these panels should be provided in a supplementary figure.*

The sequence of these substrates are/were provided in Supplementary Table 3 of the current and previous manuscript.

- *14. Fig. 3d. This graph gives the impression that RppH reacts well with double-stranded 5' ends when, on the contrary, it has previously been shown to react much more poorly with base-paired 5' ends than with unpaired 5' ends.*

We did not directly compare the substrate specificity of RppH since this is not at the center of this manuscript that rather deals with NUDT2. However, we can clearly see the phosphatase activity of RppH on dsRNA. To avoid confusion regarding RppH substrate specificity, we opted to remove RppH from Fig. 3c (previous Fig. 3d).

- *15. Fig. 4a, 4b, 4c, 4d. What is the basis for concluding that the high-mobility spot near the top of the TLC plates is phosphate (Pi) and not pyrophosphate? The detection of diphosphorylated RNA products in Fig. 4e, 4f, 4g, 4i, and 4j is consistent with the release of phosphate. However, the main product of the reaction of RppH with triphosphorylated RNA was previously shown to be pyrophosphate, yet in Fig. 4d the product of that reaction is roughly aligned with the calibration mark for phosphate (Pi). Is it possible that the conditions used for TLC were unable to separate phosphate from pyrophosphate? Marker lanes containing radioactive phosphate and pyrophosphate should be included in the TLC experiments to clear up this uncertainty.*

We thank reviewer 3 for this knowledgeable comment. We were not able to discriminate phosphate from pyrophosphates in the TLC experiments. For that reason, we reverted to HPLC experiments that show a release of single phosphates after NUDT2 treatment. Since we can identify gradual dephosphorylation of the dinucleotides that were used as a substrate, we suggest that NUDT2 cleaves off single phosphates. We agree with reviewer 3 that it would be interesting to study phosphate release from PPP-RNA by RppH. However, we feel that this is not at the center of this manuscript, which is focussing on NUDT2, and we ask reviewer 3 for his/her understanding that we did not repeat the TLC experiments to include additional markers for the RppH cleavage experiment. To avoid any doubt on the integrity of the spot, we labeled the TLC experiments with gP rather than Pi.

- *16. Lines 567–568. The statement that “NUDT2 is highly conserved throughout evolution” should be toned down, as Supplementary Fig. 2b compares its sequence only in worms, flies, and vertebrates.*

As suggested, we toned down this sentence. It now says, “Notably, NUDT2 is conserved throughout evolution, which suggests that this form of antiviral immunity may represent an ancient antiviral defense mechanism.”

- 17. Lines 570–571. Change “mechanisms to RNA decay pathways” to “mechanisms to evade RNA decay pathways.”

Sentence changed.

- 18. Lines 579–580. The mechanism by which miR-122 binding would protect the 5' end of HCV RNA from attack by NUDT2 requires more explanation in light of the authors' conclusion that this enzyme reacts well with base-paired 5' ends.

We apologize that this was not clearly stated in the manuscript. We did not claim miR-122 impairs NUDT2 activity but cite literature that states that miR-122 generally stabilizes HCV RNA. It may well be that the association of miR-122 to the RNA prevents efficient cleavage of the RNA by XRN-I or any other nuclease. We do not believe that miR-122 influences NUDT2 and do not intend to study the influence of this microRNA on NUDT2's activity.

- 19. Line 587. The acronym PRR needs to be defined.

We updated the text.

- 20. Line 844, Fig. 5a. Although the legend states that “one of three biological experiments” is shown, what actually is shown are the results of triplicate experiments in vitro with purified enzymes.

This is true. We updated the figure legend accordingly.

Reviewer Figure 1

adapted from Binder and Sulaimanov et al., 2013, PLoS Pathog.

Reviewer Figure 1

(a) Timecourse of in-vitro transcribed RNAs, electroporated into Hep3B cells, bearing different 5'-modifications (m7G-, PPP-, or P-RNA). Each RNA was co-electroporated with capped RNA as an internal control. Each set of co-electroporated RNAs is shown next to each other. The RPLP0-normalized ΔCt values were further normalized to the respective ΔCt values of capped RNA at time-point 0.5h. **(b)** The same data as in (a) is displayed, normalized to the Ct value of the co-electroporated capped RNA, and further normalized to ΔCt of the sample with capped RNA at time-point 0.5h. **(c)** Stability of a replication-deficient HCV RNA mutant (NS5B Δ GDD) in highly permissive cells and less permissive cells, respectively. Half-life-time is estimated to be 0.92h. Adapted and permitted by Binder and Sulaimanov *et al.*, "Replication Vesicles are Load- and Choke-Points in the Hepatitis C Virus Lifecycle", PLoS Pathog, 2013

Reviewers' comments:

Reviewer #1 (Remarks to the Author):

My concerns have been adequately addressed

Reviewer #2 (Remarks to the Author):

Although the manuscript by Laudenbach et al. presents an interesting possibility where Nudt2 could act as a phosphatase that can control the fate of triphosphorylated RNA in vitro and in cells, unfortunately, the authors did not address the original shortcomings of the manuscript. It is not evident that the in vitro or the RNA transfection assays are tested in the linear range of the assays making the interpretation questionable.

1. In response to point #1 where there was a concern that the 3 hr in vitro assays time point was too long to obtain reliable conclusions, the authors respond by adding a new supplementary figure (Figure S1c) and state the following:

“We can clearly see that the assay does not give reliable results after 1h and is not saturated after 3h of NUDT2 incubation with RNA. We, therefore, opted to use a 3h incubation of RNA with NUDT2 for optimal performance of this assay.”

Supplementary Figure 1c does not support their conclusion and further validates my original concern. Examination of the values at 300nm and 600nm concentration of protein used at the 3 hr and 6 hr time points do not yield the expected 2 fold difference in activity. The differences are not considerably different. In addition, comparison of phosphatase activity between the 3 hr and 6 hr time points also reveals minor differences. Both results demonstrate the assay saturated at least by 3 hrs. In contrast, examination of the values at 1hr, show the expected ~2fold difference between the 300 and 600nm concentration of protein used and thus seems an appropriate time frame to carry out the assays. The need for using shorter time points is further supported by the “Reviewer Fig 1a” provided by the authors. Differences are detected between monophosphate RNA and triphosphate RNA with the 0.5 and 1 h times, but not the longer times.

My original concerns with the timing of the in vitro assays still stands and all the assays should be carried out at shorter times (~1hr), not 3 hrs as presented in the manuscript.

2. (original Points #2 & #3): The authors again maintained the 3-4 hr reaction times in figures 3 and 5 and my original concerns as reiterated above still stand.

3. (original Points #4) In response to my concern that the 4 and 6 hr time points in Fig 5c are not reliable, the authors present a reviewer figure to support their position that the time frame is valid. Interestingly, the data presented to rebut my concern in fact further supports it. In “Reviewer Fig 1c” the authors use figures from the Kaderali lab that detected a half life of 0.92hr for replication deficient HCV RNA in cells. Even using this time frame, one can estimate <5% of the input RNA would be left in the 4-6hr time

frame, which is basically background levels of RNA that does not provide any insight into RNA half-life. Values to obtain reliable half-lives should be derived during time points around the RNA half life of a given RNA (~1h in this case). The fact that the detected RNA does not appear to change from 4 to 6 hours further reinforces the artifactual nature of these data. It would be very surprising for transfected RNAs without base modification and/or protective 3' polyA tail to have half-lives in the 4-6 hr range.

Reviewer #3 (Remarks to the Author):

The revised manuscript addresses a number of issues raised about the previous version, but the colorimetric assays used to monitor phosphate release by NUDT2 *in vitro* remain a major concern. Multiple observations demonstrate the inadequacy of these assays and suggest that they were allowed to proceed too far toward completion (3 hr incubation with 600 nM NUDT2) to draw meaningful conclusions about the RNA substrate preferences of NUDT2, which are the entire point of Figure 3:

- (1) Substituting alanine for a critical glutamate residue in the active site of NUDT2 (E58A) appears from the colorimetric assays to have only a 2-3 fold effect on the catalytic activity of the enzyme (Fig 2b, 2f, 3a). The magnitude of this effect is miniscule in comparison to what has previously been observed for a similar mutation in other Nudix hydrolases.
- (2) That NUDT2 actually is almost completely inactivated by the E58A substitution is clearly evident in the assays shown in Fig 4, where the conversion of substrate to product was monitored by radiolabeling or reverse-phase HPLC instead of by colorimetry.
- (3) Contrary to definitive evidence that RppH requires unpaired nucleotides at the 5' end of its RNA substrates, the colorimetric assay gives the impression that base-paired 5' ends react as well with RppH as with the nonspecific phosphatase CIP (lines 472-473 and Fig 2b).

The authors have done two additional experiments to address the reviewers' concerns about their data, but neither is persuasive.

- (1) In an effort to bolster confidence in the colorimetric assays, they have added a figure panel (Suppl Fig 1c) in which they compare phosphate release by NUDT2 at two enzyme concentrations (300 nM and 600 nM) as a function of time. However, this experiment serves only to reinforce the aforementioned concern that these assays were allowed to proceed too far toward completion. As judged by colorimetry, 600 nM NUDT2 generates about twice as much product as 300 nM NUDT2 after 1 hr (as expected), but this ratio declines substantially at later times (3 hr, 6 hr) as the reaction begins to approach completion. Unfortunately, the authors have used a 3 hr incubation with 600 nM NUDT2 as their standard reaction condition.
- (2) To address the possibility of a contaminating phosphatase activity in NUDT2 purified from *E. coli*, NUDT2 and the NUDT2 E58A mutant were each immunoprecipitated from HEK293T cells and then assayed colorimetrically. The revised text (lines 475-476) states that, when isolated in this manner, NUDT2 but not NUDT2 E58A released phosphates from RNA. However, the purported inactivity of the mutant enzyme preparation is contradicted by the graph in Fig 2f.

In sum, two key tests that should have been negative controls, NUDT2 E58A and the reactivity of RppH with a base-paired 5' end, instead detected enzyme activity nearly equal to that of wild-type NUDT2, outcomes that undermine confidence in the reliability of the colorimetric assay. Unfortunately, the two additional experiments included in the revised manuscript fail to restore confidence in this assay.

Dear Reviewer #2 and #3,

We thank you and the reviewers for their rapid response, yet we were a bit surprised by the decision. We thought to have adequately addressed the reviewer's concerns in the first revision and apologize for having misinterpreted some of the points raised. We highly appreciate the chance to address these points once more, which we feel improved the manuscript substantially. Our new data addresses the raised concerns and again underlines that NUDT2 can dephosphorylate PPP-RNA to prepare this RNA for further degradation by XRN1. Furthermore, we demonstrate the role of NUDT2 in antiviral innate immunity.

The main concern of reviewers #2 and #3 was related to the *in vitro* phosphate release assay—which has been performed for screening purposes (previous Fig. 3a)—and was, for that reason, backed up with additional experiments. The used Malachite green assay measures accumulated phosphate release at a single time point. We now repeated the majority of experiments using an EnzChek assay that allows a time-resolved analysis of phosphate release. This assay (i) confirmed the ability of NUDT2 to dephosphorylate all PPP-RNAs tested, (ii) allowed time-resolved titration of NUDT2 and confirmed that the used concentration gives an optimal signal-to-noise ratio while still measuring in the linear range of the enzymatic reaction, and (iii) shows that other Nudix proteins or mutated NUDT2 do not release considerable amounts of phosphates. We replaced all figures in question with the new assay, now providing much clearer results, all in agreement with each other.

The scope of this manuscript was to identify NUDT2 as an antiviral protein targeting PPP-RNA. From screening the activity of all relevant Nudix phosphohydrolases, we identified NUDT2 as a critical enzyme to fulfill this function and provide unquestionable evidence that this protein is important for this function *in vitro* (previous Fig. 4, 5, now Fig. 2, 4, 5). We demonstrate the influence of NUDT2 on the replication of PPP-RNA-dependent viruses (Fig. 1, 6) and furthermore generated a knockout mouse that lacks NUDT2 to perform experiments on *ex vivo* primary cells (Fig. 6). We feel that this is substantial data supporting the function of NUDT2 in dephosphorylating PPP-RNA and its involvement in the antiviral immune response.

We want to emphasize that after the groundbreaking original discovery of RppH as an mRNA processing enzyme in *B. subtilis* and *E. coli*, several follow up studies characterized the precise mechanism of this enzyme (Deana, Celesnik and Belasco, 2008) (Bessman et al., 2001) (Foley et al., 2015). Similarly, we believe that our discovery of NUDT2 being involved in antiviral immunity will further stimulate research in this area.

We ask the reviewers to consider the biological relevance and impact of these findings and kindly request to re-evaluate your decision based on the points outlined below.

With kind regards,

Andreas Pichlmair

Reviewer #2:

We apologize that we misunderstood the concerns of reviewer #2 in the first round of revision. We interpreted that the primary concern of reviewer #2 was related to potentially co-purified contaminants, and for that reason, the reviewer suggested shorter time points. We addressed this potential issue in the revision by thorough purity checks using mass spectrometry (Supplementary Fig. 2b). We, additionally, purified NUDT2 from eukaryotic cells, showing similar activities as observed for bacteria-derived NUDT2 (Supplementary Fig. 4b, c).

However, this new interpretation of this concern prompted us to establish a superior assay that addresses this point. We now replaced the majority of phosphate release assays with an assay that allows time-resolved evaluation of the dephosphorylation of PPP-RNA by NUDT2 on a minute scale (EnzChek assay). We titrated the enzyme concentrations to make sure to measure in the linear range of the enzymatic reaction and re-generated all data with the optimized assay.

We feel that this new assay addresses the reviewer's concerns and concurrently validates our previous findings. We hope that reviewer #2 accepts our apologies for having misunderstood the previous concerns.

In the revised manuscript, we now present more sensitive and robust time-resolved data showing...

- (i) the dephosphorylation of PPP-RNA in a minute-resolution over a time-course of 2 h
- (ii) that reliable results are already obtained at early timepoints
- (iii) the titration of NUDT2. We identified the best signal-to-noise ratio for 600 nM NUDT2.

The new EnzChek assay strengthens our findings presented in the previous version of the manuscript: NUDT2 has phosphatase activities against PPP-RNA and is active on a broad range of substrates. Moreover, the new experiments validated the experimental settings used before, i.e., the TLC and HPLC experiments.

“Although the manuscript by Laudenbach et al. presents an interesting possibility where Nudt2 could act as a phosphatase that can control the fate of triphosphorylated RNA in vitro and in cells, unfortunately, the authors did not address the original shortcomings of the manuscript. It is not evident that the in vitro or the RNA transfection assays are tested in the linear range of the assays making the interpretation questionable.”

- (1) *“Supplementary Figure 1c does not support their conclusion and further validates my original concern. Examination of the values at 300nm and 600nm concentration of protein used at the 3 hr and 6 hr time points do not yield the expected 2 fold difference in activity. The differences are not considerably different. In addition, comparison of phosphatase activity between the 3 hr and 6 hr time points also reveals minor differences. Both results demonstrate the assay saturated at least by 3 hrs. In contrast, examination of the values at 1hr, show the expected ~2fold difference between the 300 and 600nm concentration of protein used and thus seems an appropriate time frame to carry out the assays.”*

The now used EnzChek assay enables time-resolved analysis of phosphate release during the RNA processing. We re-generated the data after providing proof that results

from 600 nM NUDT2 are in the linear range (Supplementary Fig. 3a) of the enzymatic reaction. Also, we clearly see the expected 2-fold difference in phosphate release between 300 and 600 nM NUDT2.

- (2) *“The need for using shorter time points is further supported by the “Reviewer Fig 1a” provided by the authors. Differences are detected between monophosphate RNA and triphosphate RNA with the 0.5 and 1 h times, but not the longer times.”*

In the previous version of the manuscript, we also provided Fig. 5d, measured at 1 h and 4 h post-transfection, and saw effects of NUDT2 depletion that are in line with the involvement of NUDT2 in RNA degradation. We are happy to remove the 4 h time point but believe that this timepoint further strengthens the manuscript precisely because we can reliably measure RNA even at late time points and yield highly comparable results

- (3) *“My original concerns with the timing of the in vitro assays still stands and all the assays should be carried out at shorter times (~1hr), not 3 hrs as presented in the manuscript.”*

As requested by reviewer #2, we now provide experiments that were performed up to 120 min, with single measurements every three minutes.

2. *“(original Points #2 & #3): The authors again maintained the 3-4 hr reaction times in figures 3 and 5 and my original concerns as reiterated above still stand.”*

Please see our answer above concerning the time-resolved experiment we now provide. We feel that the new optimized assay addresses these points.

3. *“(original Points #4) In response to my concern that the 4 and 6 hr time points in Fig 5c are not reliable, the authors present a reviewer figure to support their position that the time frame is valid. Interestingly, the data presented to rebut my concern in fact further supports it. In “Reviewer Fig 1c” the authors use figures from the Kaderali lab that detected a half life of 0.92hr for replication deficient HCV RNA in cells. Even using this time frame, one can estimate <5% of the input RNA would be left in the 4-6hr time frame, which is basically background levels of RNA that does not provide any insight into RNA half-life. Values to obtain reliable half-lives should be derived during time points around the RNA half live of a given RNA (~1h in this case). The fact that the detected RNA does not appear to change from 4 to 6 hours further reinforces the artifactual nature of these data. It would be very surprising for transfected RNAs without base modification and/or protective 3' polyA tail to have half-lives in the 4-6 hr range.”*

We indeed have used polyadenylated RNA in the indicated experiment. Moreover, we added additional experiments at 1 h, and 4 h post-transfection (Fig. 5d) to support the figure in question. We agree with Reviewer 2 that a time kinetic would be necessary when determining the RNA half-life. However, we were not requested to do so in the first round of revision and our calibration experiments—to establish the qPCR procedure for this RNA template—show that we can reliably detect transfected RNA at the chosen time points, evident by the detection between 19.3 and 27.2 RT-qPCR cycles, with the background detection limit being 35 cycles. We could remove Fig. 5c but feel that this additional dataset supports the functionality of NUDT2. Please note that current Fig. 5d shows similar data on RNA stability in transfected cells (supporting results of Fig. 5c), which was not disputed by the reviewers and analyzed at 1 h and 4 h post-transfection.

Given the cumulative evidence from the updated phosphate release-, TLC and HPLC-assays, which are now consistent and the RNA degradation assays in cells, we feel that we can confidently propose an involvement of NUDT2 in antiviral immunity through its ability to dephosphorylate PPP-RNA.

Reviewer #3:

“The revised manuscript addresses a number of issues raised about the previous version, but the colorimetric assays used to monitor phosphate release by NUDT2 in vitro remain a major concern. Multiple observations demonstrate the inadequacy of these assays and suggest that they were allowed to proceed too far toward completion (3 hr incubation with 600 nM NUDT2) to draw meaningful conclusions about the RNA substrate preferences of NUDT2, which are the entire point of Figure 3.”

We appreciate reviewer #3's concern. Guided by these helpful comments, we replaced the assay in question with a different assay that is more robust and allows a better signal-to-noise ratio (EnzChek assay). This assay also enables a time-resolved quantification of phosphate release from PPP-RNA while the Nudix enzymes are processing the RNA. We furthermore, provide proof that performing the EnzChek experiment with 600 nM NUDT2 produces data in the linear range of the assay (Supplementary Fig. 3a).

- (1) *“Substituting alanine for a critical glutamate residue in the active site of NUDT2 (E58A) appears from the colorimetric assays to have only a 2-3 fold effect on the catalytic activity of the enzyme (Fig 2b, 2f, 3a). The magnitude of this effect is miniscule in comparison to what has previously been observed for a similar mutation in other Nudix hydrolases”*

While the previous assay showed a partial (but highly significant) reduction in phosphatase activity, the new assay indicates no phosphatase activity of the NUDT2 mutants in this experimental setup, which is in line with the HPLC and TLC assays provided in Fig. 2 and Fig. 4.

- (2) *“That NUDT2 actually is almost completely inactivated by the E58A substitution is clearly evident in the assays shown in Fig 4, where the conversion of substrate to product was monitored by radiolabeling or reverse-phase HPLC instead of by colorimetry.”*

Using the new assay (EnzChek), the obtained data (Fig. 3a, c) is now in compliance with the HPLC and TLC experiments.

- (3) *“Contrary to definitive evidence that RppH requires unpaired nucleotides at the 5' end of its RNA substrates, the colorimetric assay gives the impression that base-paired 5' ends react as well with RppH as with the nonspecific phosphatase CIP (lines 472-473 and Fig. 2b).”*

The new experiments succeed in reproducing data proofing the inactivity of RppH on base-paired RNA constructs (Supplementary Fig. 3c, IVT4) and the activity on unpaired substrates (Fig. 3c, A4(AGAA)). In the new version of the manuscript, we now address the difference in activity and mode of action of RppH as a pyrophosphatase in comparison NUDT2. We cannot explain why RppH showed unanticipated results in the

previously shown assay. After contacting NEB about this issue, we were unable to reproduce the previous findings with a new lot of RppH. Thus, we had to proceed on the assumption that our previous data was based on an artifact specific to this batch of RppH. The new assay, performed with the new LOT, is in line with the literature, and we cannot see dephosphorylation of PPP-dsRNA, as expected.

“The authors have done two additional experiments to address the reviewers' concerns about their data, but neither is persuasive.

(1) In an effort to bolster confidence in the colorimetric assays, they have added a figure panel (Suppl Fig 1c) in which they compare phosphate release by NUDT2 at two enzyme concentrations (300 nM and 600 nM) as a function of time. However, this experiment serves only to reinforce the aforementioned concern that these assays were allowed to proceed too far toward completion. As judged by colorimetry, 600 nM NUDT2 generates about twice as much product as 300 nM NUDT2 after 1 hr (as expected), but this ratio declines substantially at later times (3 hr, 6 hr) as the reaction begins to approach completion. Unfortunately, the authors have used a 3 hr incubation with 600 nM NUDT2 as their standard reaction condition.”

The new assay now allowed us to precisely address early time points of this reaction and to quantify the phosphate release every three minutes. We are providing proof that the concentration of 600 nM NUDT2 yields results in the linear range of the enzymatic reaction (Supplementary Fig. 3a). We believe that this new assay significantly improved the manuscript, and we thus replaced most *in vitro* phosphatase experiments with the new assay.

(2) “To address the possibility of a contaminating phosphatase activity in NUDT2 purified from E. coli, NUDT2 and the NUDT2 E58A mutant were each immunoprecipitated from HEK293T cells and then assayed colorimetrically. The revised text (lines 475-476) states that, when isolated in this manner, NUDT2 but not NUDT2 E58A released phosphates from RNA. However, the purported inactivity of the mutant enzyme preparation is contradicted by the graph in Fig 2f.”

We apologize for the inaccurate wording—reviewer #3 is right that the controls display low activity. However, this activity is similarly low for control precipitations (NS1, mock) and NUDT2 E58A but much higher for NUDT2. The new text says, “NUDT2 precipitated from HEK293T cells could release significantly more phosphates from a PPP-RNA compared to NUDT2 E58A mutant or control proteins [...]” which we feel is accurately describing the results. NUDT2 E58A and the used negative controls have some remaining activity in the Malachite green assay. It is, however, evident that the activity of precipitated NUDT2 is by far exceeding the activity of NUDT2 E58A and the other controls. Due to the experimental protocol required to isolate the proteins from HEK293T cells, this experiment could not be performed with the EnzChek assay.

“In sum, two key tests that should have been negative controls, NUDT2 E58A and the reactivity of RppH with a base-paired 5' end, instead detected enzyme activity nearly equal to that of wild-type NUDT2, outcomes that undermine confidence in the reliability of the colorimetric assay. Unfortunately, the two additional experiments included in the revised manuscript fail to restore confidence in this assay.”

We agree that the Malachite green assay was a subpar choice for our purpose, though it allowed to detect significant differences between wt and mutant NUDT2. However, we exchanged all

relevant figures with the new time-resolved EnzChek assay (Fig. 3a, c, Supplementary Fig. 3a-c). The results are now in compliance with the TLC and HPLC experiments (Fig. 2c-e). Now all figures clearly show that NUDT2 E58A does not facilitate phosphate release of PPP-RNA (Fig. 3a, b, and Supplementary Fig. 3a, b).

The new experiments also confirm the inactivity of RppH on 5' base-paired RNA (IVT4) and the activity on 5' unpaired substrate (Supplementary Fig. 3c, A4(AGAA)).

The main finding of this manuscript is the involvement of NUDT2 in antiviral immunity. We agree that additional experiments would be required to determine the exact substrate specificity, yet, this is outside the scope of our manuscript. Moreover, we took the concerns of reviewers #2 and #3 regarding the colorimetric assay very seriously and exchanged all affected figures. We believe that the discovery of NUDT2 as a mammalian protein with close functional homology to a bacterial protein is highly relevant and supported by all provided experiments. We are convinced that this work will spark additional research in this area and will further enlighten the biochemical details which remain to be answered.

References:

Bessman, M. J. et al. (2001) 'The gene *ygdP*, associated with the invasiveness of *Escherichia coli* K1, designates a Nudix hydrolase, Orf176, active on adenosine (5')-pentaphospho-(5')-adenosine (Ap5A)', *The Journal of biological chemistry*, 276(41), pp. 37834–37838.

Deana, A., Celesnik, H. and Belasco, J. G. (2008) 'The bacterial enzyme RppH triggers messenger RNA degradation by 5' pyrophosphate removal', *Nature*, 451(7176), pp. 355–358.

Foley, P. L. et al. (2015) 'Specificity and evolutionary conservation of the *Escherichia coli* RNA pyrophosphohydrolase RppH', *The Journal of biological chemistry*, 290(15), pp. 9478–9486.

REVIEWER COMMENTS

Reviewer #2 (Remarks to the Author):

In this new version of the manuscript the authors present new data to support their claim that Nudt2 is a major pppRNA phosphatase protein both in vitro and in cells and functions to control viral RNA. The authors provide new data to show the phosphatase activity of recombinant Nudt2 protein by EnzChek assay and checked the purity recombinant Nudt2 protein using mass spectrometry. However, my concerns regarding the low level of detected phosphatase activity and its potential biological function remain.

1. In response to the concern about the timeframe used in the in vitro assays, the authors used EnzChek assay which enables time-resolved analysis of phosphate release. Although phosphate release is detected, 600nM of protein is required. It is important to put this into context with a well-established RNA triphosphate RNA phosphatase protein, RppH, at equal molar concentrations to obtain a true understanding of the level of detected activity. Also, if one wants to claim biological significance of this activity, it is important to put this into context with the established function of Nudt2 on the Ap4A dinucleotide which can serve as a signaling molecule. Considering the published KM and kcat of Nudt2 on Ap4A, it is reasonable to extrapolate that the activity of Nudt2 on Ap4A is in the order of thousands of fold, if not millions of fold higher than that of pppRNA. This raises questions of whether accumulation of Ap4A dinucleotide might account for any in vivo perturbations detected.

2. Supp Fig 5e: Nudt2 ^{-/-} cells significantly accumulate Ap4A dinucleotide. It is not clear whether the addition of Ap4A to the culture media or the lipofection transfection of Ap4A reported in Sup Fig 5e will significantly change the intracellular level. What is the evidence that Ap4A is taken up by cells when added to the media? To my knowledge, dinucleotides are not efficiently taken up. As for the lipofection experiment, again, what is the evidence that an amount that would be sufficient to cause a physiological change is transfected? The fact that both conditions appear to result in the same outcome of slowed viral growth (sup fig 5e), is surprising and confusing since the Ap4A addition to the media would be expected to have very little, if any uptake (and no effect) while the lipid mediated approach should at least be more efficient.

3. Fig 5c: My original concerns about this figure still stands and I believe they are simply detecting background levels of RNA following degradation of the majority of the RNA and not detecting intact RNA. It is difficult to conceive the stability of a transfected RNA does not change from 4 to 6 hours. Also, from this figure, the authors conclude that Nudt2 selectively works on viral RNA. Just because they added sequences from the HCV onto the chimeric RNA, does not mean it works on viral RNA. Do they see a different result if they transfected a chimeric RNA with a mammalian sequence?

4. Fig 6e: it is difficult to interpret these images and conclude that there is an increased cytopathic effect in Nudt2^{-/-} cells. This is further supported by the lack of growth differences between Nudt2^{+/+} and

Nudt2^{-/-} cells (sup fig 6a) and no difference in viral RNA (sup fig 6b). Although the authors conclude this latter finding “suggest that there are additional proteins with redundant functions are operative on the organism level.”, an equally plausible explanation is that Nudt2 does not directly regulate VSV RNA function.

5. Sup Fig 2b is mentioned in the letter, but not referenced in the manuscript.

Reviewer #3 (Remarks to the Author):

The manuscript has been significantly improved by replacing the malachite green assay with the EnzChek assay.

Recommended text changes:

1. Line 81. RNase J should be described here as a 5' exonuclease. Its minor endonuclease activity is not affected by phosphate removal from the RNA 5' end.
2. Line 499. The statement that “NUDT2 mediated phosphate release required hydrolase activity” is a tautology and should therefore be modified.
3. Line 506. “RNA templates” should be changed to “RNA substrates”.
4. Fig. 5a, 5b. The RNase used in these two experiments should be identified.
5. Lines 551-552, Fig. 5c. Readers should be told whether the in vitro transcribed HCV-luciferase RNA with which the HeLa cells were transfected was triphosphorylated or capped.
6. Lines 589-590. This sentence is incomplete.
7. Lines 602-604. The text states that the accumulation of infectious virus particles increased 10- to 100-fold in NUDT2^{-/-} bone marrow supernatants, but Fig. 6b suggests that the increase was actually 10- to 30-fold.
8. Line 638. The figure that shows the preference of NUDT2 for longer RNA substrates should be identified here, as this finding is not mentioned elsewhere.
9. Line 650. The statement that NUDT2 is “conserved throughout evolution” needs to be toned down, as the sequence alignment in Supplementary Fig. 2c shows only that it is conserved in vertebrates, flies, and worms.

10. Lines 667-669. The statement here that the activity of NUDT2, like that of XRN1, may be inhibited by steric hindrance seems inconsistent with the conclusion on lines 512-514 that NUDT2 dephosphorylates PPP-RNAs irrespective of 5'-terminal base pairing.

REVIEWER COMMENTS

Reviewer #2 (Remarks to the Author):

In this new version of the manuscript the authors present new data to support their claim that Nudt2 is a major pppRNA phosphatase protein both in vitro and in cells and functions to control viral RNA. The authors provide new data to show the phosphatase activity of recombinant Nudt2 protein by EnzChek assay and checked the purity recombinant Nudt2 protein using mass spectrometry. However, my concerns regarding the low level of detected phosphatase activity and its potential biological function remain.

- 1. In response to the concern about the timeframe used in the in vitro assays, the authors used EnzChek assay which enables time-resolved analysis of phosphate release. Although phosphate release is detected, 600nM of protein is required. It is important to put this into context with a well-established RNA triphosphate RNA phosphatase protein, RppH, at equal molar concentrations to obtain a true understanding of the level of detected activity. Also, if one wants to claim biological significance of this activity, it is important to put this into context with the established function of Nudt2 on the Ap4A dinucleotide which can serve as a signaling molecule. Considering the published K_M and k_{cat} of Nudt2 on Ap4A, it is reasonable to extrapolate that the activity of Nudt2 on Ap4A is in the order of thousands of fold, if not millions of fold higher than that of pppRNA. This raises questions of whether accumulation of Ap4A dinucleotide might account for any in vivo perturbations detected.*

We established the in vitro phosphate release assay to be optimal and not saturated at a 600 nM concentration of NUDT2. However, while we demonstrated that lower concentrations of NUDT2 are also active (Supplementary Figure 3a), we continued to use 600 nM to be in compliance with the TLC and HPLC experiments (Fig. 2 and Fig. 4) and to be consistent within the manuscript.

Reviewer 2 asks for a direct comparison between NUDT2 and RppH, however, this comparison is difficult as both enzymes have different modes of action. NUDT2 mediates a consecutive release of phosphates from PPP-RNA while RppH releases diphosphates. Moreover, the substrate specificity is also very different: While RppH is active on single-stranded RNA substrates, it cannot dephosphorylate blunt double-stranded RNAs, which is often found in the context of virus infections and a prerequisite for activation of pattern recognition receptors.

On a technical side, it is difficult for us to compare the molarity between NUDT2 and RppH, as commercial RppH is provided in Units/mL, which is substrate-specific. Based on protein measurements and activity assays we estimated that approximately 360 nM RppH corresponds to 300 nM NUDT2 in respect to their activity (comparing Sup. Fig. 3a, c).

We respectfully disagree with the reviewer's opinion on the vastly different activity of NUDT2 towards AP4A and PPP-RNA. It seems far-fetched to extrapolate a hugely different activity of NUDT2 towards AP4A and different phosphorylated RNAs. To properly compare the activity of NUDT2 towards AP4A and diverse phosphorylated RNAs, we would need to perform proper comparative studies with the present reagents in controlled conditions. However, this is complicated by the completely different nature of the experiments required to test AP4A cleavage (e.g. HPLC resolved cleavage in

ATP and AMP) and RNA dephosphorylation (e.g. chemical conversion of a substrate by a released phosphate followed by a colorimetric readout). A direct comparison is therefore technically not sensible and we would prefer to omit speculations on the difference between NUDT2 activity towards AP4A and phosphorylated RNAs. Despite we agree with reviewer 2 that such experiments would be interesting, they would not change the message of this manuscript nor would they change the interpretation of any results presented.

Regarding the concern of reviewer 2 that AP4A may perturb in vivo situations: We clearly show in Supplementary Figure 5 that AP4A addition and lipofection has an antiviral effect which is opposite of the effect of NUDT2 depletion (as shown in many figures of the manuscript). We are therefore confident that the effects seen in NUDT2 knockout cells are not due to AP4A accumulation.

- 2. Supp Fig 5e: Nudt2 -/- cells significantly accumulate Ap4A dinucleotide. It is not clear whether the addition of Ap4A to the culture media or the lipofection transfection of Ap4A reported in Sup Fig 5e will significantly change the intracellular level. What is the evidence that Ap4A is taken up by cells when added to the media? To my knowledge, dinucleotides are not efficiently taken up. As for the lipofection experiment, again, what is the evidence that an amount that would be sufficient to cause a physiological change is transfected? The fact that both conditions appear to result in the same outcome of slowed viral growth (sup fig 5e), is surprising and confusing since the Ap4A addition to the media would be expected to have very little, if any uptake (and no effect) while the lipid mediated approach should at least be more efficient.*

We thank the reviewer for pointing this out. Transfection of dinucleotides is commonly used when studying the activity of dinucleotides (e.g. for cGAMP transfection to activate the STING pathway). The delivery of dinucleotides is well established and we added a reference to underline this (Jin et al. 2011). Larger amounts of these stimuli are also active, likely due to passive uptake (Guerra et al. 2020). Indeed, Ap4A can be bound and subsequently be internalized in a two-step reaction; ATP-independent-binding and ATP-dependent internalization (shown on bovine aortic endothelial cells, as Ap4A can induce vasodilation when applied by intravenous injection) (Hilderman and Fairbank 1999). However, we are actively studying the function of AP4A, where we find activity of this molecule on intra- and extracellular receptors. By applying AP4A through lipofection and simple addition we cover different potential activities that AP4A may have on cells.

We ask reviewer 2 for understanding, that we were unable to measure the intracellular concentration of AP4A after transfection and to compare this in the different NUDT2 deficient cell lines. In this manuscript, we show that NUDT2 depletion leads to a specific effect against PPP-RNA viruses. If NUDT2 depletion would result in an overall effect leading to an altered immune response we would also expect effects on other viruses (such as +RNA virus SFV and the DNA virus HSV-1). We, therefore, do not believe that AP4A accumulation has a dramatic effect on the innate immune system in our experimental settings.

3. *Fig 5c: My original concerns about this figure still stands and I believe they are simply detecting background levels of RNA following degradation of the majority of the RNA and not detecting intact RNA. It is difficult to conceive the stability of a transfected RNA does not change from 4 to 6 hours. Also, from this figure, the authors conclude that Nudt2 selectively works on viral RNA. Just because they added sequences from the HCV onto the chimeric RNA, does not mean it works on viral RNA. Do they see a different result if they transfected a chimeric RNA with a mammalian sequence?*

We agree with the reviewer on toning down the selectivity on viral RNA for these experiments. The selective processing of viral RNA was concluded from the combined results of the manuscript, as the manuscript focuses on NUDT2 enabling the degradation of PPP-RNA (specific for certain viral RNAs). But the wording was changed in this paragraph.

In compliance with other work, qPCRs can be performed on electroporated non-replicating viral RNA within our selected time-frames (Binder et al. 2013). Based on our experience and the data we provide we can conclude not to detect background RNA levels. Moreover, it would be an extraordinary coincidence if we only measured this higher “background” in NUDT2^{-/-} cells. Moreover, we added additional experiments at 1 h, and 4 h post-transfection (Fig. 5d) to support the figure in question demonstrating the prolonged life of PPP-RNA, which may not have been considered by reviewer 2. All these data show higher stability of PPP-RNA in NUDT2^{-/-} as compared to control cells and no effect for other RNAs in the same cell lines.

At this stage we cannot exclude that specific viral sequences are affecting the efficacy of NUDT2 dephosphorylation. Such effects are known for some sequences (e.g. GC rich sequences required for ZAP, poly-U required for TLR7, AT-rich elements serving as promoters for Polymerase-II to generate RIG-I ligands, and secondary structures inhibiting the activity of IFIT2). However, it is not possible to test such activities within this manuscript. The experiment in question simply states that the activity of NUDT2 is not inhibited by the viral sequences within the construct.

4. *Fig 6e: it is difficult to interpret these images and conclude that there is an increased cytopathic effect in Nudt2^{-/-} cells. This is further supported by the lack of growth differences between Nudt2^{+/+} and Nudt2^{-/-} cells (sup fig 6a) and no difference in viral RNA (sup fig 6b). Although the authors conclude this latter finding “suggest that there are additional proteins with redundant functions are operative on the organism level.”, an equally plausible explanation is that Nudt2 does not directly regulate VSV RNA function.*

Apparently, there is a misunderstanding about Sup. Fig. 6a as this is a survival curve of KO mice. Therefore, we think that the morphological phenotype in vitro cannot be directly compared to vivo data.

We want to emphasize that there is a clear difference in the shape of infected wt and Nudt2^{-/-} cells in vitro, with the clear cytopathic effects in Nudt2^{-/-} cells. In addition to the images (Fig. 6e), this phenotype is in line with higher virus growth in the same Nudt2^{-/-} cells (Fig. 6c). Also, the higher virus load of NUDT2 KO is in line with the rest of the independent experiments of the manuscript.

5. *Sup Fig 2b is mentioned in the letter, but not referenced in the manuscript.*
We thank reviewer 2 for pointing this out. We updated the manuscript.

Reviewer #3 (Remarks to the Author):

The manuscript has been significantly improved by replacing the malachite green assay with the EnzChek assay.

We thank reviewer 3 for the comprehensive feedback and added changes in the text based on these recommendations.

Recommended text changes:

1. *Line 81. RNase J should be described here as a 5' exonuclease. Its minor endonuclease activity is not affected by phosphate removal from the RNA 5' end.*

Changed in the manuscript

2. *Line 499. The statement that “NUDT2 mediated phosphate release required hydrolase activity” is a tautology and should therefore be modified.*

Changed in the manuscript

3. *Line 506. “RNA templates” should be changed to “RNA substrates”.*

Changed in the manuscript

4. *Fig. 5a, 5b. The RNase used in these two experiments should be identified.*

Changed in the manuscript

5. *Lines 551-552, Fig. 5c. Readers should be told whether the in vitro transcribed HCV-luciferase RNA with which the HeLa cells were transfected was triphosphorylated or capped.*

Changed in the manuscript

6. *Lines 589-590. This sentence is incomplete.*

Changed in the manuscript

7. *Lines 602-604. The text states that the accumulation of infectious virus particles increased 10- to 100-fold in NUDT2^{-/-} bone marrow supernatants, but Fig. 6b suggests that the increase was actually 10- to 30-fold.*

Changed in the manuscript

8. *Line 638. The figure that shows the preference of NUDT2 for longer RNA substrates should be identified here, as this finding is not mentioned elsewhere.*

Changed in the manuscript

9. *Line 650. The statement that NUDT2 is “conserved throughout evolution” needs to be toned down, as the sequence alignment in Supplementary Fig. 2c shows only that it is conserved in vertebrates, flies, and worms.*

Changed in the manuscript

10. *Lines 667-669. The statement here that the activity of NUDT2, like that of XRN1, may be inhibited by steric hindrance seems inconsistent with the conclusion on lines 512-514 that NUDT2 dephosphorylates PPP-RNAs irrespective of 5'-terminal base pairing.*

We agree with reviewer 3. It has been shown that viruses evolved methods to escape the surveillance of some innate immune sensors (e.g. RIG-I, IFIT1) to escape their specific activity. We simply want to raise the possibility that similar effects may exist for NUDT2.

References:

Binder, Marco, Nurgazy Sulaimanov, Diana Clausnitzer, Manuel Schulze, Christian M. Hüber, Simon M. Lenz, Johannes P. Schlöder, et al. 2013. “Replication Vesicles Are Load- and Choke-Points in the Hepatitis C Virus Lifecycle.” *PLoS Pathogens* 9 (8): e1003561.

Guerra, J., A-L Valadao, D. Vlachakis, K. Polak, I. K. Vila, C. Taffoni, T. Prabakaran, et al. 2020. “Lysyl-tRNA Synthetase Produces Diadenosine Tetrphosphate to Curb STING-Dependent Inflammation.” *Science Advances* 6 (21): eaax3333.

Hilderman, R. H., and A. T. Fairbank. 1999. “Binding and Internalization of p1,p4-Diadenosine 5'-Tetrphosphate by Bovine Aortic Endothelial Cells.” *Biochimie* 81 (3): 255–60.

Jin, Lei, Krista K. Hill, Holly Filak, Jennifer Mogan, Heather Knowles, Bicheng Zhang, Anne-Laure Perraud, John C. Cambier, and Laurel L. Lenz. 2011. “MPYS Is Required for IFN Response Factor 3 Activation and Type I IFN Production in the Response of Cultured Phagocytes to Bacterial Second Messengers Cyclic-Di-AMP and Cyclic-Di-GMP.” *The Journal of Immunology* 187 (5): 2595–2601.

REVIEWERS' COMMENTS

Reviewer #2 (Remarks to the Author):

The authors have adequately addressed my concerns.